# A broad analysis of splicing regulation in yeast using a large library of synthetic introns

**Dvir Schirman**[1], **Zohar Yakhini**[2,3], **Yitzhak Pilpel**[1]*, **Orna Dahan**[1]

**1** Department of Molecular Genetics, Weizmann Institute of Science, Rehovot, Israel, **2** School of Computer Science, Herzliya Interdisciplinary Center, Herzliya, Israel, **3** Computer Science Department, Technion, Haifa, Israel

* pilpel@weizmann.ac.il

**Data Availability Statement:** All relevant raw files sequencing data are available in the Sequence Read Archive. Accession number PRJNA631112. All the scripts used for data analysis and for producing the figures for this manuscript can be found in https://github.com/DvirSchirman/

## Abstract

RNA splicing is a key process in eukaryotic gene expression, in which an intron is spliced out of a pre-mRNA molecule to eventually produce a mature mRNA. Most intron-containing genes are constitutively spliced, hence efficient splicing of an intron is crucial for efficient regulation of gene expression. Here we use a large synthetic oligo library of ~20,000 variants to explore how different intronic sequence features affect splicing efficiency and mRNA expression levels in *S. cerevisiae*. Introns are defined by three functional sites, the 5' donor site, the branch site, and the 3' acceptor site. Using a combinatorial design of synthetic introns, we demonstrate how non-consensus splice site sequences in each of these sites affect splicing efficiency. We then show that *S. cerevisiae* splicing machinery tends to select alternative 3' splice sites downstream of the original site, and we suggest that this tendency created a selective pressure, leading to the avoidance of cryptic splice site motifs near introns' 3' ends. We further use natural intronic sequences from other yeast species, whose splicing machineries have diverged to various extents, to show how intron architectures in the various species have been adapted to the organism's splicing machinery. We suggest that the observed tendency for cryptic splicing is a result of a loss of a specific splicing factor, U2AF1. Lastly, we show that synthetic sequences containing two introns give rise to alternative RNA isoforms in *S. cerevisiae*, demonstrating that merely a synthetic fusion of two introns might be suffice to facilitate alternative splicing in yeast. Our study reveals novel mechanisms by which introns are shaped in evolution to allow cells to regulate their transcriptome. In addition, it provides a valuable resource to study the regulation of constitutive and alternative splicing in a model organism.

## Author summary

RNA splicing is a process in which parts of a new pre-mRNA are spliced out of the mRNA molecule to produce eventually a mature mRNA. Those RNA segments that are spliced out are termed introns, and they are found in most genes in eukaryotic organisms. Hence regulation of this process has a major role in the control of gene expression.

The budding yeast *S. cerevisiae* is a popular model organism for eukaryotic cell biology, but in terms of splicing it differs, as it has only few intron-containing genes. Nevertheless,

SplicingLib Other supporting data are included within the manuscript's Supporting information Files.

**Funding:** Project costs in this work were supported by the "The Minerva Center for Live Emulation of Evolution in the Lab". YP is a Kimmel Investigator at the Weizmann Institute. The funders had no role in study design, data collection and analysis, decision to publish, or preparation of the manuscript.

**Competing interests:** The authors have declared that no competing interests exist.

this species has been used to study basic principles of splicing regulation based on its ~300 introns. Here we used the technology of a large synthetic genetic library to introduce many new intron-containing genes to the yeast genome, to explore splicing regulation at a wider scope than was possible so far.

Reassuringly, our results confirm known regulatory mechanisms, and further expand our understanding of splicing regulation, specifically how the yeast splicing machinery interacts with the end of introns, and how through evolution introns have evolved to avoid unwanted misidentifications of this end. We further demonstrate the potential of the yeast splicing machinery to alternatively splice a two-intron gene, which is common in other eukaryotes but rare in yeast. Our work presents a first-of-its-kind resource for the systematic study of splicing in live cells.

## Introduction

RNA splicing has a major role in eukaryotic gene expression. During splicing, introns are removed from a pre-RNA molecule towards creation of a mature and functional mRNA. In human, splicing is central to gene expression, as a typical gene contains 8 introns [1], and these introns can be alternatively spliced to create different alternative isoforms, with a potential to also contribute to proteomic diversity [2–5]. However, most introns are constitutively spliced [6,7] and their contribution to gene expression is not through increasing proteome diversity. Nevertheless, because a pre-mRNA must undergo splicing to produce a functional mRNA, the efficiency of this process directly affects the efficiency of the overall gene expression process. Hence, regulation of constitutive splicing can be a mechanism to regulate gene expression [8] and a target for evolution to act on [9].

The budding yeast *Saccharomyces cerevisiae*, like other *hemiascomycetous* fungi, has a low number of intron-containing genes compared to other eukaryotes [10]. Most of these genes have a single intron which is constitutively spliced. Yet, although they occupy a small part of the genome, these intron-containing genes are highly expressed and are common among cellular functions such as ribosomal genes [11]. Hence, yeast cells tightly regulate the splicing process.

Recent investigations of splicing efficiency in yeast focused on analyzing natural introns in the genome, by using RNA-seq data [12–14], by studying a library based on a single reporter gene containing natural introns from the *S. cerevisiae* genome [15], or by investigating intron sequence features and examining the evolution and conservation of natural introns [16,17]. Other studies have utilized large synthetic libraries to study how sequence features affect alternative splicing decisions [18–21] in other eukaryotes.

In this work, we systematically study cis-regulatory features that affect splicing efficiency of an intron by using a large synthetic oligonucleotide library. This enables a much larger scale exploration of intron features compared to existing studies in yeast. In addition, as opposed to previous library based studies of splicing, the present library is mostly focused on constitutive splicing regulation. This technique, based on on-array synthesis [22], was used previously to explore different elements involved in regulation of transcription, translation, RNA stability, and other regulatory elements [23–28]. Here we designed and synthesized a library of approximately 20,000 oligos, each consisting of a unique intronic sequence, either synthetic or natural, aimed to explore a range of sequence determinants that may affect and regulate splicing efficiency. We then measured splicing efficiency of each oligo using targeted RNA sequencing. Using this oligo library, we cover and explore many sequence features that can affect splicing, expanding far beyond the repertoire of natural introns.

Introns are defined by three sequence elements, the 5' donor site (5'SS), the branch site (BS), and the 3' acceptor site (3'SS). The mechanisms of 5'SS and BS recognition by the splicing machinery in *S. cerevisiae* are well understood [29]. However, the exact mechanism of interaction of the spliceosome with the 3'SS is not yet fully understood, as *S. cerevisiae* lacks a splicing factor which is present and crucial for 3'SS recognition in higher eukaryotes (U2AF1) [30]. Hence it was suggested that the 3'SS is recognized through a scanning mechanism and that any HAG (i.e. [A/C/T]AG) site could be recognized [31]. However, a vast majority of 3'SS of *S. cerevisiae* natural introns are in fact YAG (i.e. [C/T]AG), just like observed in higher eukaryotes [32].

Working with a synthetic library of introns allows us to also study the evolution of introns and their splicing. In particular, natural introns from 11 different yeast species were incorporated in our library design. This enabled us to examine how intron architecture co-evolves with changes in the splicing machinery. Specifically, we compare how the *S. cerevisiae* splicing machinery splices introns coming from species with or without the U2AF1 splicing factor encoded in their genomes. Additionally, we show that *S. cerevisiae* has a tendency to select alternative downstream 3' splice sites and produce cryptic splice isoforms. We suggest that this is related to the loss of U2AF1, and that it has shaped the evolution of the intron architecture.

Lastly, we examine the potential of the budding yeast to feature alternative splicing. In *S. cerevisiae*, a vast majority of intron-containing genes have a single intron, and there are only a handful of examples of regulated alternative splicing in this organism [33–37]. We examine the extent to which the splicing machinery produces multiple splice isoforms when given a new synthetic two-intron gene. Meaning, how easy it is for *S. cerevisiae* to produce alternative splicing if the necessary information is embedded within genes?

Reassuringly, results from our high throughput synthetic system confirm previous results on intronic *cis*-regulatory sequence elements, and alternative splicing obtained from the yeast endogenous intron-containing genes. Using the power of a large synthetic library this work expands our understanding on 3'SS selection, and on how alternative 3'SS usage in yeast has shaped intron architecture. We also provide much more comprehensive evidence for the capacity of the yeast splicing machinery to alternatively splice two-intron genes. A different aspect of our work is that it provides a resource, largest of its kind, of thousands of intron-containing mRNAs associated with splicing efficiencies, which can be harnessed to further study splicing regulation in a well-studied model organism.

## Results

### High-throughput splicing efficiency measurements of thousands of synthetic introns

To explore how the intron architecture affects splicing efficiency, we designed a synthetic oligonucleotide library [22] of 18,705 variants. All the oligonucleotides were cloned into the same location inside a synthetic gene that was then integrated into the yeast genome. The synthetic gene that includes the library was designed to reduce translation to a minimum, to avoid any differences between variants that might result from differences in translation. The synthetic background gene was based on *MUD1*, an endogenous intron-containing gene from *S. cerevisiae* genome, and translation was reduced by mutating all ATG codons in all reading frames (see alignment between the synthetic background gene and *S. cerevisiae MUD1* in S3 Data).

Each designed oligo consisted of fixed sequences for amplification and cloning, a unique 12 nt barcode, and a 158 nt-long sequence that contained a unique intron design. The 158 nt sequence was based on the *MUD1* sequence (starting 5 nt before its intron) where the endogenous splice sites were randomly mutated to destroy the original intron. Unless stated otherwise

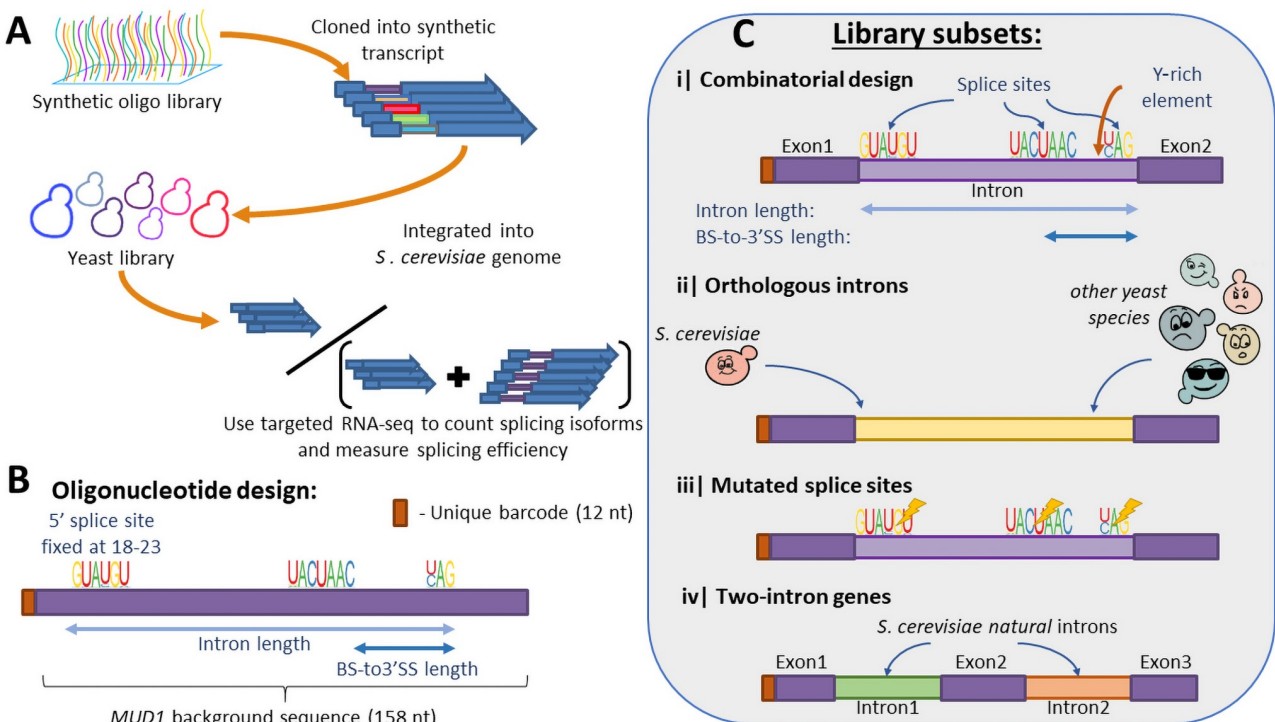

**Fig 1. A designed synthetic intron library in budding yeast. A.** A large oligonucleotide library of designed introns was synthesized and cloned into a synthetic gene. The gene was then integrated into the budding yeast *S. cerevisiae* genome, to produce a pooled yeast library. Splicing efficiency was measured using targeted RNAseq of the intronic region, identification of RNAseq reads according to a unique barcode, and of spliced isoforms using alignment of exon-intron and exon-exon junctions. **B.** Oligonucleotides design strategy—All oligos were identified using a unique 12nt random barcode at their 5' end, followed by a 158 nt long design. A 5' splice site was placed at positions 18–23 of the oligo sequence, and a branch site and 3' splice site were placed at appropriate positions according to the choice of length parameters. **C.** The library was composed of four subsets—i) Synthetic introns based on a combinatorial design representing different splice site sequences and other intronic features; ii) A set of natural introns from *S. cerevisiae* and other 10 yeast species was introduced into the library; iii) A set of synthetic introns with random mutations introduced to the consensus splice sites; iv) A set of synthetic two-intron genes produced by pairing together short intron sequences and placing an exon between them.

below, each variant was created by replacing segments of this background sequence at specific sites (for example, to introduce splice sites at desired sites). The oligo library was then cloned into the synthetic gene described above. The expression of the library's gene is driven by a synthetic promoter that was chosen from an existing promoter library [23,38] based on its high expression and low noise characteristics (see Materials & methods). After cloning the library into a vector, the entire intron-containing gene library was integrated into the YBR209W open reading frame in *S. cerevisiae* genome using a high-throughput Cre-Lox based method [39]. A schematic description of oligos structure and library creation is shown in Fig 1.

The library was composed of four major subsets, and a fifth set of negative control variants which are not expected to be spliced (see Fig 2). The first subset represents a combinatorial design introducing different splice site sequences with their exact sequence as observed in the genome, on the background of the original *MUD1* sequence. Each intron design is characterized by the sequence of its three functional sites (5'SS, BS, 3'SS), its length, the distance between its BS and 3'SS, and by the length of a short U-rich element upstream to the 3'SS. Introns were created with different lengths and different BS-to-3'SS lengths that represent the length characteristics of introns from *S. cerevisiae* non-ribosomal genes. In each oligo, an intron was created by replacing the background sequence at positions 18–23 (5 nt downstream of the unique barcode) with a 5'SS sequence, and then according to the choice of intron length,

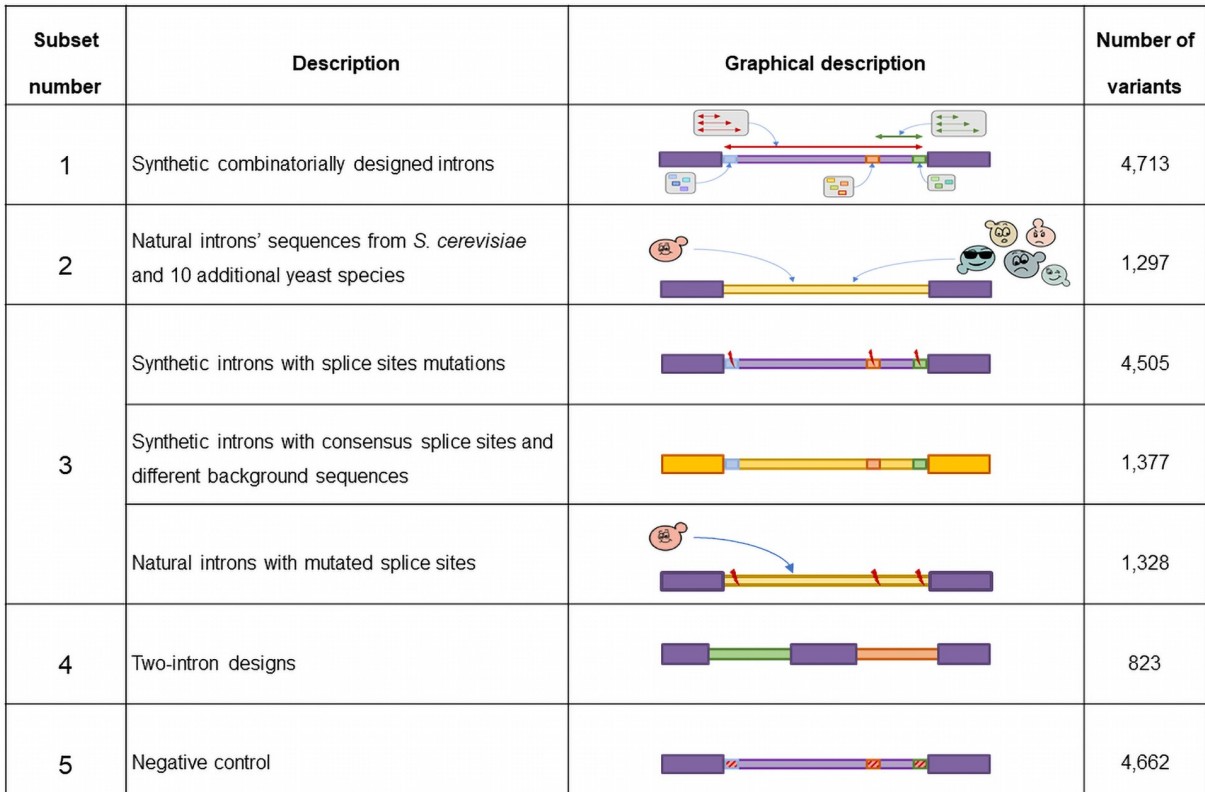

| Subset number | Description | Graphical description | Number of variants |
|---|---|---|---|
| 1 | Synthetic combinatorially designed introns | | 4,713 |
| 2 | Natural introns' sequences from *S. cerevisiae* and 10 additional yeast species | | 1,297 |
| 3 | Synthetic introns with splice sites mutations | | 4,505 |
| 3 | Synthetic introns with consensus splice sites and different background sequences | | 1,377 |
| 3 | Natural introns with mutated splice sites | | 1,328 |
| 4 | Two-intron designs | | 823 |
| 5 | Negative control | | 4,662 |

**Fig 2. Summary table of the different subsets composing the library and the number of variants in each of them.**

and BS-to-3'SS a BS sequence and a 3'SS were placed instead of the background sequence at corresponding positions (see S1 Table). In addition, three versions of short poly uracil sequence were inserted upstream to the 3'SS replacing the background sequence.

While in the first subset only the splice site sequences were taken from natural introns, a second subset of the library was composed of full intronic sequences of naturally occurring introns in *S. cerevisiae* and in other yeast species. The sequences of these introns were inserted into the 158nt synthetic oligo, replacing the existing *MUD1* background sequence at its 5' end (introns longer than 158 nucleotides were not used for this set). The third subset was based on perturbations to the two former subsets by introducing mutations to the genome-observed splice site sequences. Mutated variants were created for each of the individual three sites separately, and for all four possible combinations of the three sites. This subset also included an additional set of synthetic variants with different background sequences (instead of *MUD1* intron) that was created by introducing consensus splice site sequences at varying length properties, within different background sequences, using the same combinatorial logic used to generate the first synthetic subset. Lastly, the fourth subset of variants was composed of designs with two short introns, one next to the other, separated by a short exon, such that the entire two-intron design fits into the 158 nt oligo. The introns sequences used for this subset were natural introns taken from the *S. cerevisiae* and *S. pombe* genomes that are short enough to fit with another intron inside the 158 oligo, in addition to five short synthetic intron sequences. This last subset enables us to study the potential of the *S. cerevisiae* splicing machinery to process genes with multiple introns and to produce alternative splice variants. In addition to the above four subsets a set of negative control variants was created by introducing mock

randomly chosen sequences instead of the three splice sites, while using the same design principles as the first combinatorial subset, creating negative control variants with variable distances between the mock splice sites. The same randomly chosen mock sites were used for all the variants in this subset.

The splicing efficiency of each variant was measured using targeted (PCR based) RNA sequencing of the library's variable region. The sequence amplicon that was deep-sequenced included both the unique barcode of each intron design and its entire variable region, in either its unspliced or spliced forms. This allowed us to calculate splicing efficiency for each intron design. Shortly, each variant was identified by its unique barcode, and then the relative abundances of the unspliced and spliced isoforms were determined by aligning exon-intron and exon-exon junction sequences against the RNAseq reads. The splicing efficiency of a design was defined as the ratio between the spliced isoform abundance and total RNA abundance of this design. If for a specific RNA read, neither an unspliced isoform nor an intended spliced isoform of the designed intron were identified, we searched for an mRNA isoform that might have been a result of a novel unintended splicing event (for details see Material and methods). We note the possibility that unspliced isoforms might have higher turnover rates [40], and therefore, since we sequenced RNA in its steady state our method might overestimate splicing efficiencies, yet this effect is likely equal across all library variants.

## Synthetic introns are successfully spliced within the genomic construct

We used targeted RNA sequencing to measure splicing efficiencies, 99.43% of the library's variants were identified using this process. 13,220 variants in the library were designed with a single intron (The remaining 5,485 are negative control variants or two-intron variants). For 33.5% of these single intron variants, we observed the designed splice isoform with median splicing efficiency of 0.428 (on a scale from 0 to 1). For comparison, of the 3,347 single-intron variants that were designed to serve as negative controls (as they miss regulatory sites for splicing), only 1.64% yielded a spliced isoform and the median splicing efficiency there was 0.048. In addition, when examining the natural introns of *S. cerevisiae* that were included in our library, we saw that 84% of them yielded spliced isoforms, with median splicing efficiency of 0.675. We examined the total splicing efficiency distribution (i.e. splicing of any possible intron including cryptic introns) for all the variants that present greater than zero splicing efficiency (we chose to ignore variants with zero splicing efficiency as most of them are a result of non-functional intron designs). We observe a bi-modal distribution of splicing efficiency, when most variants are either mostly spliced, or rarely spliced (S1A Fig). This suggests that the splicing machinery acts mostly as a binary switch.

To verify that the introduction of a 12nt barcode upstream of the intron does not have a significant effect on splicing efficiency, we attached four different barcodes to each of 517 randomly chosen designs. We then computed the variance in splicing efficiency between each quartet of barcoded designs (considering only designs with non-zero splicing efficiency) and compared it to the variance obtained with random quartets of barcodes that do not belong to the same design. Reassuringly we see that the mean variance among the correct quartets is much lower than the mean variance of each of $10^4$ randomly shuffled variants quartets (S1B Fig). We further examine the pairwise correlations between splicing efficiency of pairs of library variants that share the same sequence yet different barcodes (S1C Fig). Reassuringly, we found a significant positive correlation between pairs of variants with different barcodes (mean Pearson correlation, r = 0.76). Nevertheless, we notice that in ~33% of the cases, most of them with intermediate splicing efficiency values, we observe significantly different splicing efficiency values between variants that differ only in their barcode, suggesting that there is

some effect for the barcode—a random 12 nucleotides exonic sequence, located slightly upstream to the 5' end of the intron. Yet, this result indicates that the barcode choice exerts at most a low effect on splicing efficiency. All together, these results establish the validity of our system as an *in vivo* quantitative splicing assay.

## Splicing efficiency is positively correlated with RNA abundance

When examining total RNA abundance, we see a significant positive correlation between total splicing efficiency (i.e. a sum of splicing efficiency values over all spliced isoforms for each variant) and total RNA abundance (i.e. summed level of unspliced and spliced isoforms) (Fig 3A). Since the calculation for splicing efficiency is dependent on total RNA abundance per design, we compare this result to a random set taken from the same distribution (see S1D Fig legend). No correlation was observed in the random set (S1D Fig). This observed correlation between

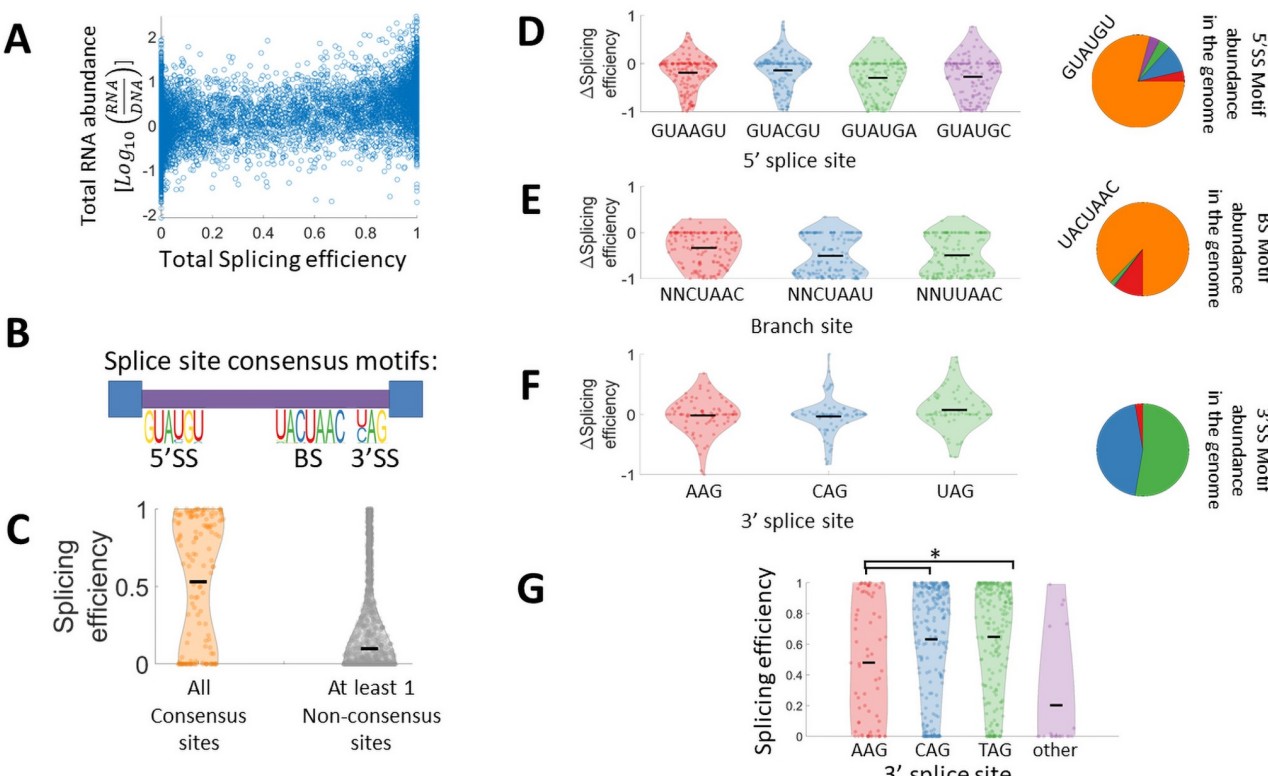

**Fig 3. A set of combinatorially designed synthetic introns elucidate splice sites' variants contribution to splicing efficiency. A.** Scatter plot of the total RNA abundance and total splicing efficiency shows a significant positive correlation between RNA level and splicing efficiency (Pearson r = 0.47 p-value$<10^{-100}$). RNA abundance is defined as the log ratio between the RNA read frequency and the DNA read frequency per variant. **B.** Splice site motifs for *S. cerevisiae* introns as determined by their frequency in *S. cerevisiae* natural introns, one can notice a dominant consensus sequence for the 5'SS and BS, and two consensus sequences for the 3'SS. **C.** Distribution of splicing efficiency for synthetic intron variants with consensus splice sites at all 3 sites (orange), is significantly higher than splicing efficiency distribution for variants with at least one non-consensus splice site (grey) (two sample t-test, p-value$<10^{-80}$). Black horizontal line represents mean splicing efficiency. **D-F.** Effect of non-consensus splice site sequences on splicing efficiency. Violin plots represent the distribution of the difference in splicing efficiency between a variant with a single non-consensus splice site, to a corresponding consensus sites variant which is identical in any other parameter. For the 3'SS since there are two consensus sequences, AAG variants were compared against the average of the two consensus variants, and CAG/UAG variants were compared against the other consensus variant. Pie charts show the relative abundance of each splice site in *S. cerevisiae* genome (orange slice represents the consensus site, and other colors correspond to the colors in the violin plots). In the case of the BS, NNCUAAC non-consensus variants represent all sequences that fit this template but different from the consensus sequence UACUAAC. Note that for the BS pie chart in (E) the blue portion representing NNCUAAU variants is too small to be visible. **G.** Splicing efficiency distribution only for the natural introns set considering introns with consensus 5'SS and BS, and binned according to different 3'SS (AAG introns' splicing efficiency are significantly lower than CAG/TAG introns, t-test p-value$<0.003$).

splicing efficiency and total RNA abundance per design is intriguing. We note that since this correlation is obtained with synthetic transcripts that were not selected in evolution to regulate their gene expression through splicing, it suggests a molecular mechanism that may be at work. Previous works have shown that unspliced intron-containing genes are degraded through the nonsense-mediated decay (NMD) machinery [41,42] which can explain this observation. This observed positive correlation might also be explained by effects of splicing on nuclear export [43], or on RNA stability [40,44], that can both enhance steady-state RNA levels per gene. Additionally, it was shown that splicing and transcription occur simultaneously [13], and it was recently demonstrated in mammalian cells that inefficient splicing of non-coding RNA can lead to transcription termination [45]. In addition, this correlation could be explained by an effect of transcription rate on splicing but we estimate that this explanation is less likely since all variants in the library are transcribed using the same promoter. Yet, it is possible that sequence variation in our library affects RNA pol II elongation rate which might in turn affect splicing efficiency.

## Combinatorial design of introns elucidates features contributing to splicing efficiency

We next analyzed the set of synthetic introns created by combinatorial design of different splice site sequences and length properties. As expected we noticed that introns that contain the consensus splice site sequence in all three splice sites are, as a population, better spliced than introns with at least one non-consensus splice site (Fig 3B and 3C). Next, for each of the non-consensus splice site variants we examined how it affects splicing efficiency by analyzing variants with a single non-consensus splice site, and comparing their splicing efficiency to the corresponding design with consensus splice sites and otherwise identical sequence (Fig 3D-F). We notice that almost all non-consensus branch site sequences result in much lower splicing efficiency, although they all contain the catalytic A residue at position 6, indicating that out of all three functional sites, the branch site is most crucial for efficient splicing (Fig 3E). On the other hand, in the 5'SS, while on average the non-consensus variants are spliced less efficiently we do observe a substantial number of variants that are spliced better than the corresponding variant with consensus site (Fig 3D), we also notice that for two of the splice site variants, lower splicing efficiency is observed only for longer introns (S2A Fig). For the 3'SS we see that there is no measurable difference in splicing efficiency between the three variants found in the genome, although two of them are significantly more abundant than the third (Fig 3F). The fact that the AAG 3'SS variants are spliced as well as the two YAG 3'SS variants is surprising due to the fact that ~95% of introns in all eukaryotes use a YAG 3'SS [32], and due to cryo-EM based structural analysis that demonstrates the chemical mechanism of interaction of the spliceosome with the -3 pyrimidine in the 3'SS [46]. However, when considering the set of natural introns with consensus splice sites at their 5'SS and BS we notice that introns that utilize AAG as their 3'SS are spliced less efficiently, suggesting that in natural intron sequences there is embedded information that disfavors AAG as a 3'SS (Fig 3G). Additionally, we used a set of variants with random mutations in their splice sites (with fixed length properties), to analyze the effect on splicing efficiency of all possible single nucleotide mutations in the three splice sites, and this analysis replicated the results observed for the splice sites variants found in the genome (S2B Fig). We have also analyzed the effect of predicted secondary structure on splicing efficiency, and whether it has a different effect depending on the splice site. We introduced mutations next to splice sites positions with consensus splice site sequences that result in a predicted stem-loop structure, where the splice site resides within the stem (all mutations were outside of the splice site sequence itself). These mutations are therefore predicted to render the

splice site inaccessible. We observe that in most cases these mutations reduce splicing efficiency significantly (paired t-test, p-value$<10^{-40}$), and we observe that different splice sites have different sensitivity to changes in RNA secondary structure, for example folded 5'SS variants are spliced better than folded branch sites variants (S2C Fig). We do note, though, that creating these predicted structures may have involved several mutations that might have affected other intron properties.

A subset of the library included variants with combinatorially designed synthetic introns as described above created on the background of 9 other sequences (instead of the *MUD1* background sequence used for most of the library) using only consensus splice sites and variable length properties. Two of these alternative sequences come from other intron-containing genes from the *S. cerevisiae* genome (*UBC9*, *SNC1*), and 7 other sequences are random sequences. By examining the difference in splicing efficiency compared to the same combinatorial design in the *MUD1* background we notice that the background sequence can have a significant effect, either increasing or decreasing splicing efficiency (S2D Fig). On average the two background sequences from natural genes result in small decrease in splicing efficiency (*UBC9*, paired t-test p-value = 0.01) to no effect (*SNC1*, paired t-test, p-value = 0.2), while all random sequences decrease splicing efficiency on average (paired t-test, p-value$<0.01$).

Next, we examined how other intron features can affect splicing efficiency. In addition to the splice sites, an intron is characterized by a poly-pyrimidine tract upstream to the 3'SS [47]. However, it was noticed that in yeast a weaker feature is observed compared to other eukaryotes, and it is characterized by short uracil-rich sequence instead of pyrimidine (U or C) [17,48]. Using our library, we examine the effect of uracil rich sequences upstream to the 3'SS, by binning all the variants that utilize consensus splice site sequences, according to their U content in a 20nt window upstream to the 3'SS, and then comparing the splicing efficiency distribution in each bin (Fig 4A). A striking pattern of correlation between higher U content in this window and increased splicing efficiency emerges. To demonstrate that the observed effect is specific to uracil enrichment and not to pyrimidine enrichment, we repeat the same analysis by binning according to Y content, considering only variants with well-balanced U and C composition. We observe no correlation between Y enrichment and splicing efficiency in this setup (S3A Fig). This result is the first experimental evidence that *S. cerevisiae* splicing machinery is specifically affected by a poly uracil tract, as opposed to other eukaryotes [47].

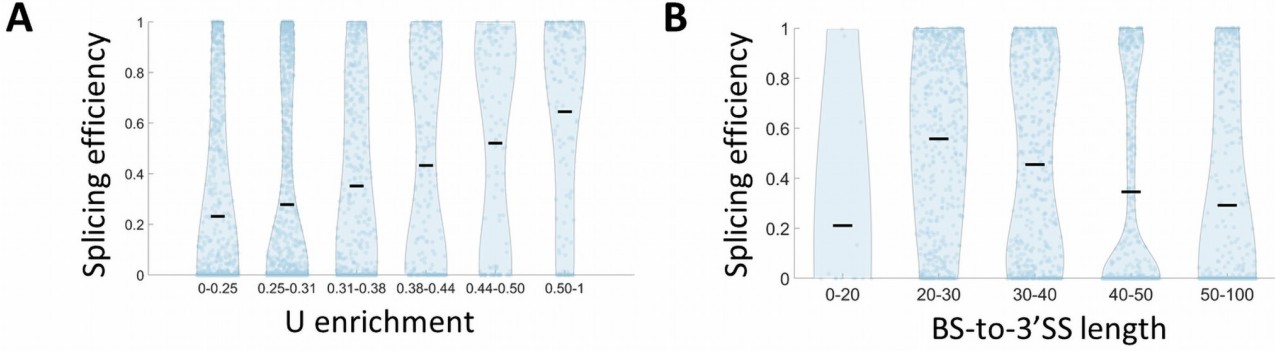

**Fig 4. BS-to-3'SS nucleotide content and length affect splicing efficiency. A.** Splicing efficiency distribution for library variants utilizing consensus splice sites' sequences, is binned according to poly uracil tract enrichment, which is calculated as the U content at a window of 20 nucleotides upstream to the 3'SS. We observe a significant positive correlation, compared to a non-significant correlation for Y-rich elements (S3A Fig) (Pearson r = 0.95 p-value = $4 \cdot 10^{-3}$). **B.** Splicing efficiency distribution for library variants utilizing consensus splice sites' sequences, is binned according to BS-to-3'SS length. We notice that splicing efficiency is significantly higher for length of 20–30 nt compared to all other bins (t-test, p-value$<10^{-4}$), and a negative correlation between splicing efficiency and BS-to-3'SS length for lengths $\geqq 20$ (Pearson correlation, r = -0.14 p-value$<10^{-10}$).

Another feature that was identified as important for efficient splicing is the distance between the BS and 3'SS [16,49,50]. This feature is also highlighted by the recent high resolution cryo-EM structural analysis of the spliceosome when it was demonstrated that at the transition between branching conformation to the exon-ligation conformation (C to C*) the BS is removed from the spliceosome catalytic core, to allow space in the active site for the docking of the 3'SS. This transition requires a minimal distance between the BS and the 3'SS [32,51–53]. By binning all the variants that utilize consensus splice site sequences according to their BS-to-3'SS length we observe that a distance of 20–30 nt is optimal in terms of splicing efficiency, and that splicing efficiency is negatively correlated with this distance for distances longer than 20 nt (Pearson correlation r = -0.14, p-value$<10^{-10}$) (Fig 4B). This result suggests a slightly longer optimal distance than the 13–22 nt distance that was observed in a previous work that used a single intron with varying 3'SS positions [50].

Previous studies have associated other intronic features with splicing efficiency such as, intron length [54,55], and intronic GC content [15,49,56,57]. The data from the current library significantly supports the effect of intronic GC content. Specifically, we observed that splicing efficiency decreases with increasing GC content (S3B Fig). As for intron length, we do not observe a specific length that is spliced more efficiently (S3C Fig). We note though that introns taken for this library were bounded by a length of 158nt and the distribution of intron lengths represented in this library represent the length distribution of introns from non-ribosomal genes in *S. cerevisiae*. Introns from ribosomal genes are longer (mean intron length of ~400 nt). These lengths are not represented in this work. Thus, we cannot exclude the possibility that intron length does affect splicing efficiency, if such dependence affects only longer introns.

## Cryptic splicing events drive intron evolution

Up to this point, we have focused on "intended" splicing events. That is, successful splicing of the designed intron of each variant in the library. However, our designed constructs might also result in cryptic splicing isoforms, different from the intended ones. To identify such splicing events, for each variant, we looked for cryptic spliced isoforms by aligning the RNAseq reads to the full unspliced sequence, allowing large gaps in the alignment. A long uninterrupted gap in the RNA read is potentially a spliced intron, and if at least one of its ends is not found at the designed ends of this variant we label it as a "cryptic intron".

We found cryptic splice isoforms in 25.2% of the variants designed to have a single intended intron, with a median splicing efficiency of 0.038 for the cryptic splice isoforms. Note that this is significantly lower than a median splicing efficiency of 0.428 for the designed splice isoforms. We then studied the location of the cryptic intron ends relative to the intended intron ends, and found that 87% of them have the same 5'SS as the designed intron, while only 1% of them have the same 3'SS (Fig 5A). Further, most cryptic splicing occurs through 3'SS sites that are downstream, but not upstream to the designated 3'SS (Fig 5A).

This observation suggests that a vast majority of cryptic splicing events are a result of utilization of the canonical 5'SS with an alternative 3'SS during the splicing process. We acknowledge the possibility that due to our amplicon sequencing based method, there is a lower chance to detect putative upstream alternative 5'SS selection, and that alternative 5'SS might possibly result in unfinished splicing intermediate product [58], which also would not be detected by our method.

Since we observe cryptic isoforms as a result of alternative 3'SS utilization we inspect how the designed 3'SS affects the levels of cryptic isoforms. We consider only variants with consensus sites in their 5'SS and BS. As was shown for the designed isoforms (Fig 3F), we don't

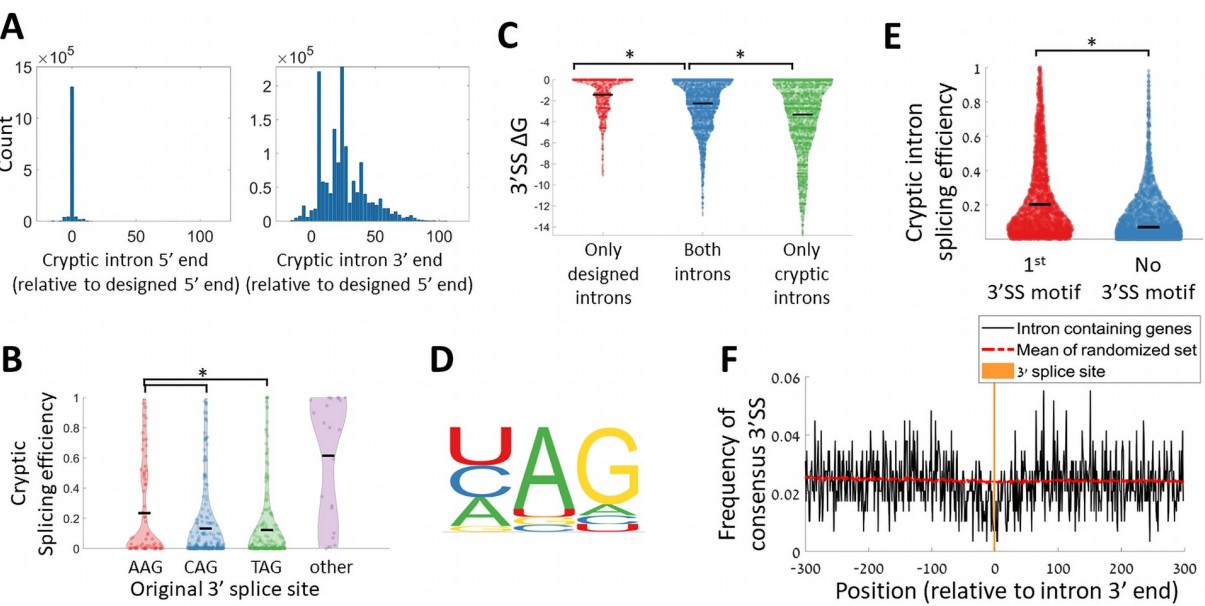

**Fig 5. Cryptic splice isoforms are produced due to selection of alternative 3'SS. A.** Relative splice site position distribution for cryptic introns (relative to the designed splice site position), for the 5' splice site (left), and the 3' splice site (right). **B.** Distribution of cryptic isoforms splicing efficiency for the set of natural introns, binned according to the original intron's 3'SS. Introns utilizing AAG 3'SS have significantly higher cryptic splicing efficiency values (t-test p-value<0.005) **C.** Distribution of predicted ΔG values at a window of 30 nucleotides around the designed 3'SS for variants with designed splice isoform and no cryptic splice isoform (red), variants with both designed and cryptic splice isoform (blue), and variants with only cryptic splice isoform (green). The difference between all three distributions is significant (t-test, p-value < $10^{-18}$). **D.** Sequence motif of the 3'SS for all cryptic intron isoforms detected. **E.** The distribution of splicing efficiency for cryptic splice isoforms, binned according to isoforms in which the cryptic intron is spliced at the first appearance of a 3'SS motif (red), or isoforms for which the last 3 nucleotides of the introns are not a 3'SS motif (blue). (t-test, p-value<$10^{-100}$). **F.** 3'SS motif avoidance pattern around introns' 3' end in the *S. cerevisiae* genome. All *S. cerevisiae* intron-containing genes were registered according to the 3' end of their intron. The black line presents the frequency of TAG/CAG motif for each position. The red dashed line presents the expected frequency by averaging the motif frequencies over 1000 sets of sequences registered according to random positions inside coding genes. Orange vertical line represents the position of the 3'SS.

observe a difference in cryptic isoforms levels between different 3'SS sequence variants, for synthetic introns (S3D Fig). Importantly, however, we do observe a significant increase in cryptic isoforms levels for natural introns utilizing AAG 3'SS as the designed 3'SS compared to the two common 3'SS sites (Fig 5B). This suggests that AAG sites are only seldomly found in the yeast genome because they may lead to higher unintended alternative 3'SS utilization.

Previous work has also found alternative 3' splice site usage events in *S. cerevisiae* and suggested that the 3'SS choice can be explained by local RNA secondary structure at the original 3'SS [59]. To examine this suggestion, using our synthetic introns data, we considered the distribution of the predicted RNA free energy (ΔG) at a window of 30 nucleotides around the designed 3'SS. We found that spliced variants with no cryptic splicing have more open predicted structures at their 3'SS compared to spliced variants with cryptic splicing, and to a greater extent than unspliced variants with cryptic introns (Fig 5C). When studying the active alternative 3'SS we see that 70% of the alternative isoforms are spliced at one of the three sequence motifs found in the genome ([U/C/A]AG) (Fig 5D) and that 68.5% of them are spliced at the first downstream occurrence of this 3'SS motif after the designed 3'SS. Those isoforms that are spliced at the first downstream 3'SS motif, are more efficiently spliced than other cryptic splice isoforms (Fig 5E).

The observation that the *S. cerevisiae* splicing machinery can easily misidentify 3'SS leads us to hypothesize that mechanisms to avoid such cryptic splicing events must exist, as these events can result in frameshifts and in premature stop codon occurrences. Hence, we checked if we

observe a selection against 3'SS motifs near the 3' end of natural introns in the genome. We registered all *S. cerevisiae* introns at their 3' end and calculated the frequency of the two dominant 3'SS motifs ([C/T]AG) around introns' end. Indeed, we found a depletion of these motifs at a window of -50 to +30 around introns' end compared to a null model based on 1000 random sets of genomic loci (Fig 5F).

## Co-evolution of the splicing machinery and intron architecture across yeast species

Our system allows us to introduce any short intron sequence into the *S. cerevisiae* genome. This gives us the opportunity to study the evolution of intron architecture by introducing introns from other yeast species and observing how well they are spliced in our system. We first introduced all the naturally occurring introns from *S. cerevisiae* genome that can fit in our oligonucleotide design length constraint. For each intron we inserted the full length of the intron flanked by 5 exonic nucleotides from each end. This sequence was inserted on the background of the standard *MUD1* derived background sequence of the library, at its 5' end. Hence, the length limit for an intron was 148 nucleotides (i.e. to fit a 158 oligo and allowing 5 exonic nucleotides in each end), amounting to 149 introns out of 299 in this species. It should be noted that this limit on intron length forces us to use only introns from non-ribosomal genes in our library, as all the introns in ribosomal genes in *S. cerevisiae* are significantly longer (mean length of ~400 nucleotides).

Next, for each natural *S. cerevisiae* intron, we included in our library introns from orthologous genes from a set of 10 other yeast species, according to orthology identified by [16]. We found that most *S. cerevisiae* endogenous introns are spliced in our system (85.5%). Interestingly, introns from most of the other species are, typically, also spliced at similar efficiencies (Fig 6A). Furthermore, we compare the splicing efficiency of each intron to its corresponding orthologous intron in *S. cerevisiae*, and define ΔSE as the difference in splicing efficiency for ortholog introns of the same gene compared to *S. cerevisiae*. We found that many of the non *S. cerevisiae* introns are spliced better than their *S. cerevisiae* orthologs (Fig 6B), suggesting that *S. cerevisiae* introns are not specifically optimized for high splicing efficiency by their own splicing machinery.

Although we did not see a specific preference for the natural introns of *S. cerevisiae*, we still observe that introns from some species like *E. cymbalariae* or *K. thermotolerans*, are spliced in lower efficiency compared to introns from other species. We further note that these two species do not stand out phylogenetically from others (Fig 6A and 6B), specifically we note that *K. lactis* is as distant phylogenetically as these strains, but its introns are spliced similarly to *S. cerevisiea's* natural introns. Hence, we hypothesized that introns from these species might have been optimized to evolutionary changes in the splicing machinery in their original species. One such molecular candidate could be the gene U2AF1, which is a splicing factor that is associated with the location of the branch site relative to the 3' end of the intron [49]. This gene is missing in 6 out of 11 of the yeast species we analyze here including *S. cerevisiae*. In additional species (e.g *T. blatae*) this factor is highly mutated and probably non-functional [16]. Indeed, the introns from the 11 yeast species we used here show a different distribution of BS-to-3'SS distances, that is concordant with the absence or presence of U2AF1 (S4A Fig), while other properties are not significantly different between the two groups (S4B-E Fig) (intron length distribution does seem to be different for the two groups, but this difference is solely ascribed to the BS-to-3'SS distance, as can be seen by the lack of difference in 5'SS-to-BS distances (S4B and S4C Fig)). When comparing the distribution of splicing efficiencies between introns from species with or without U2AF1, we observed that introns that come from species lacking

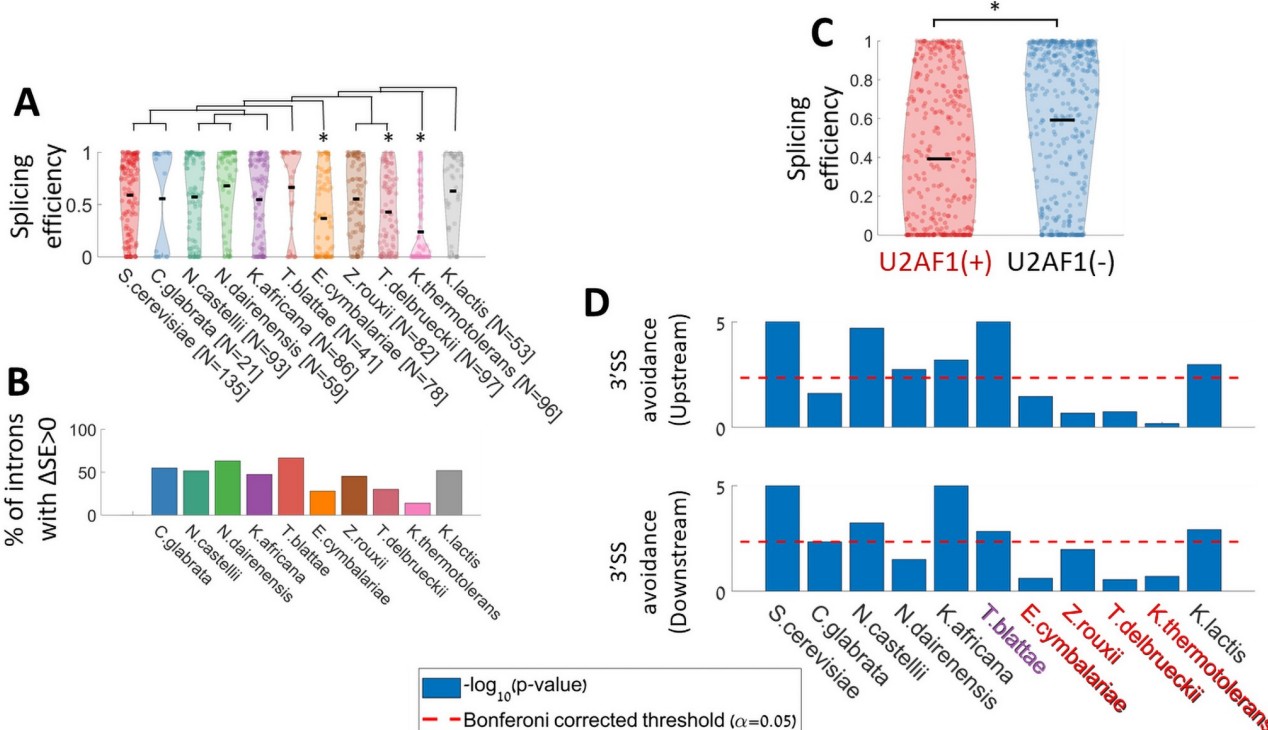

**Fig 6. Analysis of ortholog introns from other yeast species reveals intron architecture evolution. A.** Splicing efficiency distribution of spliced variants of *S. cerevisiae* natural introns and orthologs of *S. cerevisiae* intron-containing genes. A species phylogenetic tree (created according to [60]) is presented above the corresponding violins. The number of introns included in the library from each species is indicated after each species name. Asterisks mark origin species with splicing efficiency distribution significantly lower than that of *S. cerevisiae*'s introns (t-test, p value<0.001). **B.** Percent of introns that are spliced better than their *S. cerevisiae* ortholog intron for each species. **C.** Splicing efficiency distribution for introns that come from species that have a functional copy of U2AF1 splicing factor (left), and introns that come from species without U2AF1 splicing factor (t-test, p-value<$10^{-9}$). **D.** Hypothesis test for the 3'SS motif avoidance for each of the 11 species upstream (top) or downstream (bottom) to the 3'SS. P-value was calculated by comparing the mean frequency of the 3'SS motif at a 30nt window upstream/downstream to the 3'SS against $10^5$ sets of sequences each composed of coding genes sequences registered according to randomly chosen positions. Species with a copy of U2AF1 are marked in red, species with malfunctioned U2AF1 are marked in purple, and species without any copy of U2AF1 are marked in black.

U2AF1 are better spliced in our *S. cerevisiae* system, which, as noted, lacks U2AF1 (Fig 6C). Hence, we suggest that introns that were adapted to a splicing machinery that uses U2AF1 diverged in evolution and are less suitable to *S. cerevisiae* splicing machinery.

In the previous section, we demonstrated that *S. cerevisiae* has a tendency to splice cryptic introns at alternative 3'SS downstream of the original site, leading to a selection against 3'SS motifs near natural introns 3' end. Interestingly, when performing the same analysis for introns originating from the other 10 yeast species, we found in all but four species, a similar significant *S. cerevisiae*-like depletion of 3'SS sequence motifs near their introns 3'SS. This suggests an active evolutionary driven avoidance of the motif in these regions. Strikingly, the 7 species that show this depletion signal are the six species lacking U2AF1, and the one species with highly mutated copy of this factor (Fig 6D and S5 Fig). Put together, our results suggest that loss of the U2AF1 gene results in a flexible recognition of the 3'SS, which in turn generates a selective pressure to avoid 3'SS motifs near the intended 3'SS in order to avoid cryptic splicing events. On the other hand, splicing machinery that includes U2AF1 results in a more stringent 3'SS recognition mechanism, possibly due to tight constraints on the BS-to-3'SS distance.

## A computational model elucidates important features that govern splicing efficiency

In this work we created a large collection of single intron variants, with a systematic exploration of different intron design features. This wide collection of variants allows us to train a computational model that predicts splicing efficiency values from sequence features. For the purpose of this model we used the set of all single-intron variants including both synthetic and natural introns from all species examined in our library, and excluding negative control variants and few variants with features that are not well defined (N = 12,667). We trained a gradient boosting model [61,62] using a 5-fold averaging cross-validation technique [63] on randomly chosen 75% of the variants set (N = 9,500). As an input to the model, we used a set of 39 features, comprising the splice site sequences (as a categorical feature), intron length parameters, GC content, 3' U-rich element, and local secondary structure predictions at each splice site (see a full list of parameters in S2 Table). The model predictions were tested on the remaining 25% of the set of single-intron variants used for this model (N = 3,167). Predicted splicing efficiency values for the test set are reasonably well correlated with the measured splicing efficiency values (Pearson r = 0.77, Fig 7A). We note that this correlation between predicted and measured splicing efficiency values is at a similar extent as the correlation between identical variants with different barcode sequence (mean pairwise Pearson correlation r = 0.76, S1C Fig). However, we also note that comparing between multiple barcodes of the same design there is mostly agreement for cases of totally unspliced on one hand, or fully spliced variants on the other, with relatively lack of correlation in intermediate splicing efficiency values. In contrast, the correlation between predicted and measured splicing efficiencies extends throughout the dynamic range of values. For that reason, we compare between the model predictive performance and the multiple barcodes set using a categorical test, and a

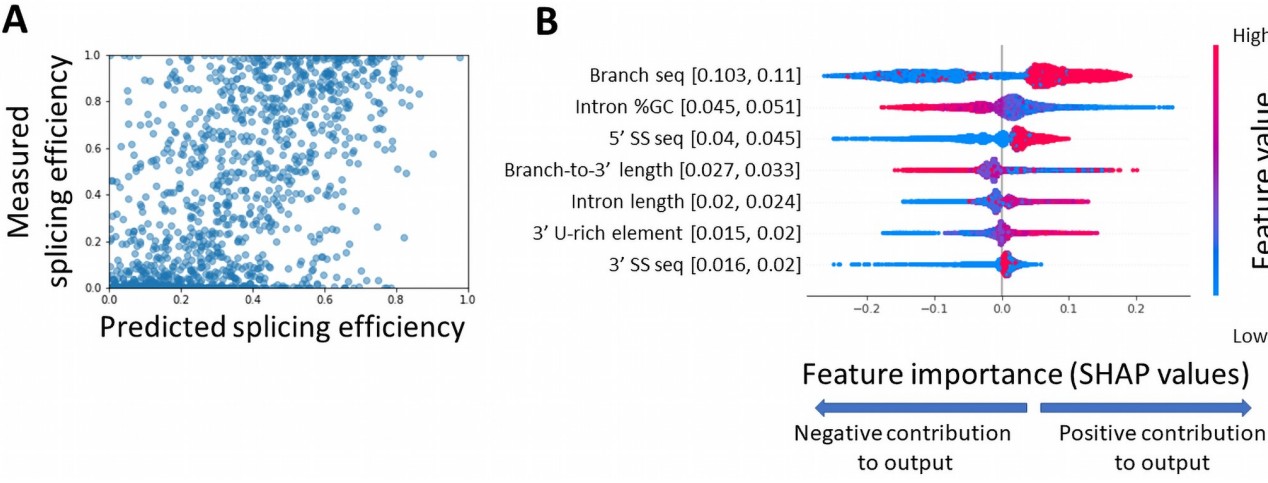

**Fig 7. Sequence features contribution to successful splicing are derived from a computational model. A.** Measured splicing efficiency values versus predicted splicing efficiency values of the test set variants (N = 3,167) as predicted by a gradient boosting model using 5-fold averaging cross-validation technique (Pearson r = 0.77 p-value$<10^{-100}$ 95% CI = [0.745, 0.774]). **B.** Distribution of Shapley values for the top 7 features when ranked according to the mean absolute value of the Shapley values. The x-axis represents the Shapley values, the higher the absolute value of the mean of the distribution, the higher its contribution to the model predictions. Positive values mean that the feature is predicted to improve splicing efficiency, and negative values mean that the feature is predicted to reduce splicing efficiency. Sample points are colored according to their feature's value for numerical features, and for splice site sequences that are treated as categorical features, they are colored by the splice site relative abundance in the *S. cerevisiae* genome (high values represent abundant sequence variants). Numbers in parentheses represent the 95% confidence interval of the mean absolute value of each feature. We notice that except for the bottom two features, the confidence intervals of features do not overlap, indicating stable ranking of the feature contributions. Confidence interval values were calculated by resampling training and test sets 1000 times.

contingency table. We categorized splicing efficiency values to three classes [unspliced, intermediate splicing, high splicing], and computed Cramer's V ($\varphi_C$) from a contingency table to derive an effect size measure for the agreement between either model predictions and measurements or between two sets of barcodes (S4 Data). The model predictions result in $\varphi_C = 0.191$, while the mean pairwise Cramer's V for the multiple barcodes comparison is $\varphi_C = 0.297$. Meaning that the model predictions are not higher than expected by the reproducibility of our dataset.

The predictive model enables us to examine the contribution of each feature to a successful prediction of splicing efficiency. We used Shapley values [64] to infer individual features' importance. Meaning, we analyzed the global contribution of each feature to the predicted splicing efficiency value across all observations. Fig 7B presents the individual feature contribution for each observation (i.e. library variant) of the 7 most important features according to this analysis, the distribution of Shapley values for each feature, and its correspondence with the feature's values. We found that the most important feature is the sequence of the BS which corresponds with the large difference in splicing efficiency we observed for non-consensus BS variants (Fig 3E). Next, we notice that intronic GC content has high contribution, as low GC content contributes to higher splicing efficiency, which is in agreement with previous findings [15,56]. The 5'SS sequence also has a high contribution to efficient splicing. Interestingly, while the 3'SS sequence is considered one of the defining features of introns, it is only ranked 7th in terms of importance for the model predictions.

## *S. cerevisiae* has the capacity to alternatively splice two tandem introns, thus generating alternative splice variants from the same RNA

Alternative splicing is not considered to have a major role in gene expression regulation in *S. cerevisiae*. There are 10 known genes with two tandem introns in the *S. cerevisiae* genome [65], and most of them are not known to be alternatively spliced. Previous works have examined alternative splicing of a two intron gene in *S. cerevisiae* by studying the spliced isoforms of the two genes that are known to be alternatively spliced (i.e. *DYN2* and *SUS1*) [36,37,66]. In these works, the regulation of alternative splicing of a specific gene was studied through chemical or genetic perturbations [37] or changing environmental conditions [36]. Here we use our library to comprehensively assess the prevalence of two-intron RNAs that can exhibit alternative splicing.

We created a subset of the library with two short introns separated by an exon. For this set we chose 25 short introns (<76 nucleotides), 10 of them are the 10 shortest natural introns in *S. cerevisiae*, additional 10 were randomly chosen from all the natural *S. pombe* introns that fit to the length limits of the library and utilize splice sites that are found in *S. cerevisiae*. Lastly, we created 5 synthetic introns with *S. cerevisiae* consensus splice sites, at a length of 56 nucleotides, and BS-to-3'SS distance of 20 nt. Using these 25 short introns, we created a set of variants composed of pairings of two introns, where the first intron was inserted at the 5' end of the variable region, and the second intron at the 3' end of the variable region, separated by an exon, the exon sequence was taken from the *MUD1* based background sequence used for other parts of the library, which resulted in a variable length of the exon depending on the length of the two introns. Altogether, this set included 823 variants.

Using this set of two-intron variants, we tested whether *S. cerevisiae* has the potential to alternatively splice, and produce multiple spliced isoforms when given a two-intron gene. Such two-intron designs can result in 5 possible isoforms (Fig 8A). For each variant, we measured the relative frequency of each of the isoforms by aligning its predicted exon-exon junctions to the RNAseq reads. We observed all 4 spliced isoforms in our data (Fig 8B).

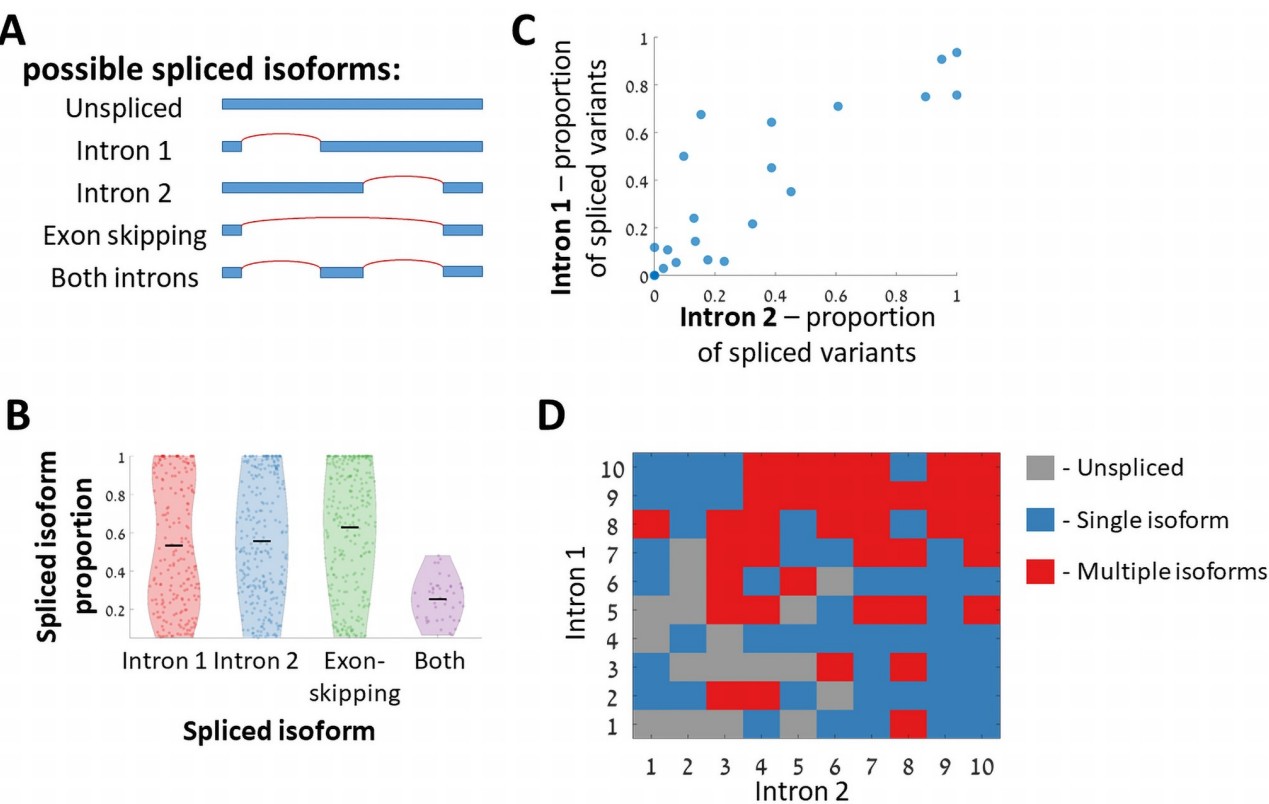

**Fig 8. Tandem two-intron designs demonstrate a capacity of *S. cerevisiae* to alternatively splice two introns. A.** Five possible isoforms can be observed for a two-intron design. **B.** Distribution of isoform's relative frequency for the 4 possible spliced isoforms. **C.** Intron performances when placed as the first intron vs. when placed as the second intron. Each dot represents a single intron sequence, the y-axis represents the proportion of variants with this intron spliced as the first intron, and x-axis represents the same for the second intron variants (Spearman correlation r = 0.88 p-value = $10^{-8}$). **D.** The number of observed isoforms for each of the natural *S. cerevisiae* intron pair variants. Each number represents a different intron sequence, and they are ordered according to the number of variants in which the intron was spliced.

Interestingly, out of 100 variants that are composed of two *S. cerevisiae* natural introns, 37 variants have more than one spliced isoform observed for the same pre-mRNA sequence. This observation suggests that for each mRNA with two such introns there is a considerable probability to create an alternatively spliced gene by combination of two introns.

We compared the relative abundance of the alternative splice variants. There were two distinct hypotheses we could test, either that in each pair of introns one of them will be better spliced than the other regardless to its location in the intron, or that the location of the two introns in the gene will dictate splicing tendency/efficiency, so that either the upstream or downstream introns will be better spliced. Our results show that the splicing efficiency of each intron is dependent mainly on its sequence, and less on the relative locations of the two introns in the gene. Isoforms with only the upstream or only downstream intron spliced, appear in similar numbers and have similar splicing efficiency distributions (Fig 8B). Further, when comparing the proportion of spliced variants for each intron sequence between variants in which it was placed as the upstream intron, and variants in which it was placed as the downstream intron, we see that those two measurements are in high agreement (Fig 8C).

To decipher which intron properties contribute to multiple spliced isoforms, we analyzed all the possible 100 pairs assembled from natural *S. cerevisiae* introns. For each of the 10 introns in this analysis we counted the number of variants for which we observed an isoform in which this intron was spliced out. Then we ranked the 10 introns according to the number

variants in which each of them was spliced (regardless of its position as the 1st or 2nd intron). We noticed that multiple isoforms are observed mainly when both introns are ranked high. A single isoform is observed when one intron is ranked low and the other high, and when a pre-mRNA consists of two introns that are ranked low, splicing is hardly observed (Fig 8D).

## Discussion

In this study we used a designed, large synthetic oligonucleotide library to study the regulation of constitutive and alternative mRNA splicing. We chose for this study the budding yeast *S. cerevisiae* because a vast majority of introns are constitutively spliced in this species, making it an ideal experimental system to study molecular mechanisms of intron splicing. However due to its low number of endogenous intron-containing genes, studies of splicing regulation and alternative splicing in this organism are based on a relatively small number of cases.

First, we observed a positive correlation between splicing efficiency and total RNA abundance. This result is in line with previous findings that incorporation of an intron often improves heterologous gene expression in *S. cerevisiae* [67] and in higher eukaryotes [68,69]. The result is nonetheless surprising to some extent in yeast since most yeast genes do not include an intron, so we would expect the gene expression machinery and its quality control to not be dependent on splicing. The fact that we observe this correlation for synthetic intron-containing genes suggests there is a molecular mechanism that mediates the correlation, as opposed to an evolutionary indirect effect. It was demonstrated that the nonsense mediated decay machinery is involved in degrading unspliced intron-containing genes [41,42], which might account for this result. However, we note that it was also shown that NMD mainly targets ribosomal intron-containing genes [41], and translating transcripts [42], while our library is based on non-ribosomal gene that was engineered to reduce translation to a minimum. Several other possible molecular mechanisms could mediate a positive effect of splicing on RNA expression levels. These include, potential effect of splicing on nuclear export [43], on RNA stability [40,44], or on transcription rate [7,13]. Conversely, we note the possibility that splicing efficiency could be increased as a result of another mechanism that in parallel increases gene expression levels. For example in Ding *et al.* [7] it was shown that splicing rates are increased due to spatial clustering of highly expressed transcripts, which in turn increases splicing efficiency through an "economy of scale" principle. However, in our experimental system we presume that is not the case since all library variants are expressed using the same promoter at the same genomic location.

Using a combinatorial design approach, we compared the effect on splicing efficiency for all naturally occurring splice site variants. Interestingly, in a set of synthetic introns we see no difference between the three possible 3'SS variants. However, when comparing full naturally occurring intron sequences we notice lower splicing efficiencies for AAG 3'SS variants. This result suggests that natural introns contain within them information that disfavors AAG 3'SS utilization. Our results further suggest that such avoidance of the AAG site might be due to the potential of this motif to increase the probability of alternative 3'SS usage. This might explain the fact that AAG 3'SS is rarely observed in fungi genomes, despite the fact that as we demonstrate here, mechanistically it can be spliced as well as the two other variants.

An intronic feature that is unique to *S. cerevisiae* and other fungi is a U-rich element at the 3' end of introns. While a pyrimidine-rich element at introns' end is very common in other eukaryotes, in *S. cerevisiae*, there is only a uracil enrichment [17]. Here, we provide the first experimental evidence that indeed U-rich elements contribute to splicing efficiency, while general Y-rich elements do not (Fig 4A). We suggest that this might reflect evolutionary changes in the yeast splicing machinery that causes it to interact specifically with uracil nucleotides.

Additionally, we provide experimental evidence that *S. cerevisiae* splicing machinery has a preferred BS-to-3'SS length (Fig 4B), which is supported by the recent high-resolution cryo-EM structural analysis of the spliceosome. These optimal lengths might indicate a linkage between the removal of the BS from the active site and the docking of the 3'SS there, although other explanations cannot be ruled out.

We used our library to study the evolution of introns architecture by introducing into it natural introns from 11 yeast species. We tested if introns that come from different hosts that use modified splicing machineries are spliced differently when processed by the *S. cerevisiae* machinery. We suggest that a loss of a splicing factor (U2AF1) that occurred in some of the yeast species examined here, affects intron architecture through the distance between the branch site and the intron's 3' end. We then observe that introns coming from hosts including this factor are spliced less efficiently in our system lacking the U2AF1 factor (Fig 6C). We suggest that introns evolved to adapt to the presence or absence of this splicing factor. Additionally, we find that the loss of this splicing factor is accompanied by a genomic depletion of the 3'SS motifs near introns' 3' ends (S5 Fig). We hypothesize that in species that lack U2AF1 splicing factor, the splicing machinery misidentifies the intended 3'SS due to similar sequence motifs near that site. This hypothesis is supported by the fact that many cases of alternative 3'SS isoforms are observed both in this current study (Fig 5A and 5B) and in previous findings in *S. cerevisiae* [35,59]. To conclude, the 3'SS in *S. cerevisiae* is recognized through a site with a low information content, and since this organism's splicing machinery lacks the structural rigidity provided by U2AF1, it has the flexibility to recognize several possible splice sites downstream of a BS. On one hand, this flexibility creates an opportunity for the organism to adapt through alternative spliced isoforms [35], but on the other hand it creates a risk of producing unwanted spliced isoforms. We believe that this observation presents a general tradeoff seen in molecular biology, when a low information motif results both in flexibility which enables adaptation, and a risk for deleterious effects which drives selection to avoid cryptic motifs. This tradeoff was also demonstrated recently for a different system, such as the bacterial promoters [70].

Interestingly, in humans, a mutation in U2AF1 was associated with hematopoietic stem cell disorders that can progress into acute myeloid leukemia, and this mutation was shown to cause missplicing events as a result to alterations in preferred 3'SS, which are presumed to be related with the disease [71–73].

Lastly, regulated alternative splicing is the focus of many studies on splicing because of its effect on increasing the proteomic diversity of the genome [18,19,21]. In *S. cerevisiae*, there are very few known examples of functional alternative splicing [33–37], and two of these examples are genes which demonstrate that the *S. cerevisiae* splicing machinery can alternatively splice a two-intron gene [36,37,66]. Here we aimed to decipher what are the *cis* regulatory elements that enable a two-intron gene to be alternatively spliced. Using a synthetic approach, we have shown that many combinations (37%) of two natural *S. cerevisiae* introns result in alternative splicing. Moreover, we have demonstrated that if each of the two introns is efficiently spliced on its own there is a very high probability for observing multiple spliced isoforms, suggesting that a combination of two introns that are designed for efficient splicing is sufficient for alternative splicing.

It is still an open question how easy it is to create a gene that is alternatively spliced in a regulated manner in response to different environmental conditions. Here we provided the first step towards answering this question by screening a set of synthetic genes for the ability to be alternatively spliced. These genes can now serve as a basis for an effort to evolve in the lab a new regulated alternatively spliced gene in *S. cerevisiae*, which might shed new light on the necessary components for regulation through alternative splicing.

## Materials & methods

### Synthetic library—General design notes

We used Agilent's oligo library synthesis technology [22] (Agilent Technologies) to produce a pool of 45,000 designed single-stranded DNA oligos at a length of 230 nucleotides. Each oligo includes two 30 nucleotides fixed homology regions at their 5' and 3' end for amplification and cloning, and a 12 nucleotide unique barcode downstream to the 5' homology. This leaves an effective variable region of 158 nucleotides for each variant. The entire synthesized library was composed of several sub-libraries aimed for different projects, these libraries were separated in the initial amplification stage using different homology sequences.

All the experiments and data reported in this paper are based on one sub-library with 18,705 variants (termed SplicingLib1).

Barcodes were chosen such that the minimal edit distance between any two barcodes will be greater than 3 to allow for single error correction for all types of errors including insertion/deletion which are the common error types in oligo synthesis.

The library was designed as a non-coding RNA library in order to avoid possible differences between variants that result from translation. Hence, for each variant, any occurrence of ATG triplet at any frame was mutated to avoid occurrences of a start codon. Except for cases where a 5'SS includes an ATG triplet, in which case, a stop codon was introduced 2 codons downstream of the ATG.

### Synthetic library—Variants design

The synthetic introns library is composed of several sets of variants. A first set is based on a combinatorial assembly of intron features. Six features were chosen to represent an intron, and all the possible combinations of features were combined to create a set of 4,713 synthetic introns. The features used for this set are: the three splice sites, 5'SS, BS, 3'SS, intron length, BS-to-3'SS length, and a 3' U-enriched sequence element. For each feature, a set of few values was chosen, the 5'SS and 3'SS sets included all the splice sites variants that are found in *S. cerevisiae* genome (5 sequence variants for the 5'SS, and 3 for the 3'SS). The BS included the consensus BS sequence (UACTAAC), and three template sequences with two random nucleotides at the first two positions, since non-consensus BS sequences differ greatly in these positions (NNCUAAC, NNCUAAU, NNUUAAC). For the intron length feature, 5 representing lengths were chosen (73, 89, 105, 121, and 137 nucleotides), and for the BS-to-3'SS length 4 representing lengths were chosen (20, 30, 40, and 50 nucleotides). For the 3' U-enriched sequence element, 3 sequences at different lengths were used (AUUUUUAA, UUUAA, UAA). In addition, for each of the splice sites, a random control sequence was created and a set of control variants was created by assembling the three control sites with all combinations of the other features (see S1 Table, for summary of the synthetic combinatorial design subset).

Full oligo sequences were based on a background sequence that was derived from the intron-containing region of *MUD1* gene from *S. cerevisiae* genome (positions 4–161 in *MUD1* open reading frame), followed by randomization of its three splice sites. Each oligo sequence was created by placing a 5'SS 5 nucleotides downstream of the effective variable region (positions 18–23 relative to the start of the barcode) instead of the background sequence in this position. Then a BS, U-rich element, and 3'SS sequences were placed in a similar manner according to the chosen length parameters of each variant. In addition, a set of 1,377 variants was created by taking only the consensus splice site sequences at different lengths and incorporating them within 9 additional background sequences, the first two from *UBC9*, and *SNC1*

(positions 34–191 and 98–255 in the gens' ORF respectively) genes in a similar manner, and the remaining 7 based on random sequences.

A second set is based on mutating consensus sites' variants from the previous set. 3,607 variants were created by introducing random mutations to the splice site sequences themselves, and 898 additional variants were created by mutating positions adjacent to splice sites with the aim to create a stem-loop RNA structure at the splice sites. This aim was achieved by introducing random mutations and selections *in-silico* of variants for which RNA secondary structure tool [74] predicts that the splice site will be base-paired within a stem-loop structure.

A third set was based on 1,297 naturally occurring introns from 11 yeast species. We first took all the endogenous intron sequences from *S. cerevisiae* [65] that fit into our 158 nucleotides effective variable region (149 introns). Each intron was inserted with a flanking region of 5 nucleotides from each side on the background of the *MUD1* derived background sequence described above. Next, we took intron sequences from orthologs of these intron-containing genes from a set of 10 other yeast species and added them to the library in the same manner. Intron annotations were taken from [16]. For the *S. cerevisiae* introns, we also created a set of 1,328 variants with random mutations in introns' splice sites. All the single-intron variants with all their features are documented in S1 Data.

Finally, a fourth set of 823 variants was created by combining two intronic sequences to create synthetic two-intron variants. For this set, we chose all the introns from *S. cerevisiae* genome shorter than 76 nucleotides (10 introns), plus 10 randomly chosen short introns from *S. pombe* genome and an additional 5 synthetic introns based on combining consensus splice sites on the background of a random sequence. Each variant sequence was created by placing two introns on the background of the *MUD1* sequence, the first intron at the 5' end of the variable region, and the second at the 3' end of the variable region. All possible pairs of intronic sequences were created and introduced to the library. All two-intron variants with all their features are documented in S2 Data.

## Construction of master plasmid

In order to integrate the library into *S. cerevisiae* genome, we used a Cre-Lox based method [39]. We built a master plasmid to clone the library into, which is compatible with this method. The master plasmid was based on pBAR3 [39]. A Lox71 site was cloned into pBAR3 to allow Cre-Lox recombination using restriction-free cloning method [75] (primers prDS20, prDS21) to create pDS101. Then we cloned into the plasmid a background sequence that will serve as the library's non-coding gene. A non-coding sequence was designed by taking the sequence of *MUD1* intron-containing gene from its transcription start site to its 3' UTR (-70 to 1106, relative to the start codon), excluding a region around the intron into which the oligo library would be cloned (-45 to 211, relative to start codon). The background sequence was then mutated at any occurrence of ATG to avoid start codons, and additional 27 sites were mutated to reduce homology to the endogenous copy of *MUD1* in the genome. In the cloning site of the oligo library two 20 nucleotides sequences were added, to be used as homology sequences during library cloning. Upstream to the background gene we added a synthetic promoter taken from a published promoter library that was chosen for its high expression level and low noise (Promoter id #2659, from Supp table 3 in [23]). Downstream to the background gene we added *ADH1* terminator sequence. The entire promoter+background gene+terminator construct (total length of 1,397 nucleotides) was synthesized as a Gene Fragment (Twist Bioscience). The synthesized background gene was cloned into pDS101 using NEBuilder HiFi DNA Assembly (New England Biolabs) to create pDS102 (primers prDS22, prDS23).

See alignment between the synthetic background gene and *S. cerevisiae MUD1* in S3 Data.

## Synthetic library—Cloning and amplification of plasmid library

Synthetic oligos were first amplified according to Agilent's recommendations [76]. Library oligos were amplified using sub-library specific homology plus 4 different 8 nucleotide sequences that were inserted to serve as an index for control purposes, such that every unique variant could be measured independently 4 times, and a homology sequence to the master plasmid for cloning.

The library was amplified in 4 PCR reactions, Each PCR reaction included:

- 25 ul—KAPA HiFi HotStart ReadyMix (Roche)

- 1.5 ul - 10uM forward primer (prDS55-58 for SplicingLib1)

- 1.5 ul - 10uM reverse primer (prDS59 for SplicingLib1)

- 200 pM of DNA oligo library

- $H_2O$ to complete volume to 50ul

    PCR program:

1. 95˚C 3 min

2. 98˚C 20 sec

3. 58˚C 15 sec

4. 72˚C 15 sec

5. Repeat steps 2–4 for 15 cycles

6. 72˚C 1 min

After amplification, the PCR product was cut from agarose gel, purified using Wizard SV Gel and PCR Clean-Up System (Promega), and all 4 reactions were pooled together. The master plasmid pDS102 was linearized using PCR reaction (primers prDS62, prDS63). Then a plasmid library was assembled using 4 independent reactions of NEBuilder HiFi DNA Assembly (New England Biolabs) to avoid biases in assembly that might affect the library's distribution.

From this stage, we followed Agilent's library cloning kit protocol [77] steps 2–7. In short, the plasmid library was purified using AMPure XP beads (Beckman Coulter), then inserted to electrocompetent *E. coli* cells (ElectroTen-Blue, Agilent Technologies) using electroporation. Then bacterial cells were inoculated into two 1 liter low gelling agarose LB bottles, in order to grow isolated colonies in 1-liter volume. After 48 hours of growth in 37˚C bacterial cells were harvested using centrifugation, and cells were grown overnight on liquid LB media in 37˚C. Finally, the amplified plasmid library was extracted from bacterial cells using 4 reactions of Midiprep kit (Macherey-Nagel NucleoBond Xtra Midi Plus).

All primers used for construction of master plasmids, cloning and amplification of oligo library can be found in S3 Table.

## Growth media

Growth media used in this work:

1. YPG - 10g/L yeast extract, 20 g/L peptone, 20 g/L galactose

2. YPD - 10g/L yeast extract, 20 g/L peptone, 20 g/L glucose

3. SC complete—6.7 g/L nitrogen base without amino acids, 20 g/L glucose, 1.5 g/L amino acid mix

4. SC -URA—6.7 g/L nitrogen base without amino acids, 20 g/L glucose, 1.5 g/L drop-out mix lacking Uracil

## Synthetic library—Integration into yeast genome

The library was inserted to *S. cerevisiae* strain yDS101 (ura3Δ ybr209w::GalCre-KanMX-1/2URA3 -lox66 HOΔ::TEF2-mCherry::pCUP1-YiFP::NAT). This strain was based on SHA185 strain, that contains Cre-Lox landing pad and is derived from BY4709 strain, kindly supplied to us by Sasha F. Levy's lab [39]. Transformation of the plasmid library to yeast cells was done using a Cre-Lox based high throughput genomic integration method [39], that inserts the plasmid sequence into the YBR209W dubious open reading frame (Chromosome II 642585). yDS101 yeast cells were transformed with 500ug plasmid library and grown overnight in YPG media to induce Cre expression. Then cells were plated on selective media (SC-Ura) approximately 50 plates per transformation. We counted the number of colony-forming units by plating diluted samples and got $1.5 \cdot 10^6$ CFUs for SPlicingLib1 which are ~60 times the number of unique variants in the library.

## Culture growth for RNA extraction

Total RNA of the library cells was extracted in two independent repeats. Library cells were grown overnight in 5 ml SC complete media in 25 ml glass tube at 30˚C on a tube roller at 200 rpm, and then diluted to a fresh media (50 ml) by 1:100 factor and grown for an additional 6 hours in a 250 ml glass Erlenmeyer flask shaken at 200 rpm until they reached $OD_{600}$ of 0.5, such that cells are harvested in mid-log phase. The cell culture was centrifuged in 15 ml plastic falcon tube for 45 seconds at 4,000g and the pellet was immediately frozen in liquid nitrogen.

## RNA extraction, cDNA synthesis, and genomic DNA extraction

RNA was extracted using MasterPure Yeast RNA Purification Kit (Lucigen), and treated with TURBO DNase (ThermoFischer) to remove any residues of genomic DNA. We then synthesized cDNA using reverse transcription with random primers using qScript Flex cDNA Synthesis Kit (QuantaBio).

In order to normalize RNA levels by the relative frequency of each variant in the sample, we extracted genomic DNA from the same samples used for RNA extraction. Cells were harvested and frozen at mid-log the same as for RNA extraction. DNA was extracted from all samples using MasterPure Yeast DNA Purification Kit (Lucigen).

## Next-generation sequencing—Library preparation

Both cDNA and genomic DNA samples were prepared for sequencing in the same manner. We used a two-step PCR protocol to amplify the library's variable region and link it to Illumina's adaptors with indexes.

The first PCR reaction was used to amplify the variable region and link homology sequences to Illumina's adaptors, we performed 8 parallel reactions to each sample to reduce PCR biases. We used 6 different forward primers each with one extra nucleotide, to create shifts of the amplicon sequence in order to avoid low complexity library.

Each reaction included:

- 25 ul—KAPA HiFi HotStart ReadyMix (Roche)

- 1.5 ul - 10uM forward primer (prDS137-142)

- 1.5 ul - 10uM reverse primer (prDS143)

- 100ng DNA

- H$_2$O to complete volume to 50ul

  PCR program:

1. 95˚C 3 min

2. 98˚C 20 sec

3. 58˚C 15 sec

4. 72˚C 15 sec

5. Repeat steps 2–4 for 20 cycles

6. 72˚C 1 min

Next, we pooled all 8 reactions for each sample and purified the PCR product using AMPure XP beads (Beckman Coulter). The second PCR was used to link specific indexes to each sample so we can multiplex several samples in a single sequencing run.
Each reaction included:

- 25 ul—Phusion High-Fidelity PCR Master Mix with HF Buffer (New England Biolabs)

- 2.5 ul - 10uM forward primer (prDS144)

- 2.5 ul - 10uM reverse primer (prDS145)

- 1-5ng DNA

- H$_2$O to complete volume to 50ul

  PCR program:

1. 98˚C 30 sec

2. 98˚C 10 sec

3. 62˚C 20 sec

4. 72˚C 15 sec

5. Repeat steps 2–4 for 15 cycles

6. 72˚C 5 min

All primers can be found in S3 Table.
Next, we purified the PCR product using AMPure XP beads (Beckman Coulter), quantified final concentration using Qubit dsDNA HS (ThermoFischer), diluted all samples to 4nM, and pooled together all the samples. NGS library was sequenced in Illumina NextSeq 500 system, using 150x2 paired-end sequencing. We obtained a total of 13.9, 12 million reads for the two RNA samples of SplicingLib1, and 1.9 million reads for the two corresponding DNA samples.

## Mapping sequencing reads to the library's variants

Sequencing reads from all samples were processed the following way: We first merged paired-end reads using PEAR [78], next we trimmed homology sequences and demultiplexed the

reads according to the 4 control indexes using Cutadapt [79], then we clustered all unique reads using 'vsearch—derep_prefix' [80].

All the unique reads were mapped to a library variant according to the first 12 nucleotides in the read, which are the designed barcode. A read was mapped to one of the library's variants by searching the barcode with minimal edit distance to the read's barcode. If this minimal distance was <3, and only a single library barcode is found in this distance, the read was aligned to this variant. Variants with less than 10 reads were discarded from the analysis.

### Data analysis

All data analysis except the gradient boosting model were done in Matlab (R2018b). Gradient boosting modeling was done in Python 3.7.

### Computing splicing efficiencies

For each variant, the mapped reads obtained from the RNA sequencing were first classified into three possible types: unspliced, intended spliced isoform, and undetermined. A read was classified into one of these types using an alignment of 40 nucleotide sequences representing *exon-intron*, and *exon-exon* junction sequences. We aligned each read to the reference junction sequences using local Smith-Waterman alignment (swalign function in Matlab), and a normalized alignment score was defined the following:

$$Junction\ alignment\ score = \frac{SW(junction, read)}{SW(junction, junction)}$$

If the normalized score was >0.8 we infer the junction is positively aligned to the RNA read.

A read was classified as unspliced if it was aligned to the two *exon-intron* junctions and not aligned to the *exon-exon* junction. A read was classified as 'intended spliced isoform' if it was aligned to the *exon-exon* junction and not to the two reference *exon-intron* junctions. All other reads were classified as undetermined.

Intended splicing efficiency for each variant was then calculated for each index according to:

$$SE = \frac{spliced\ isoform\ abundance}{total\ RNA\ abundance}$$

A final splicing efficiency value for each variant was then set by taking the weighted mean between repeats in each index, followed by taking the median between indices.

The undetermined reads were further analyzed to search for cryptic spliced isoforms, meaning, isoforms that result from splicing of an intron different than the designed intron, hence no *exon-exon* reference junction could be defined. Each read was aligned against the full reference design with the following parameters to the Smith-Waterman algorithm (Gapopen = 100, ExtendGap = 1) in order to allow for alignment with long uninterrupted gaps, if the normalized alignment score was <0.7 and the number of mismatches in the alignment was <6, a read was set as cryptic spliced isoform. Then the 5' and 3' end of the intron were set according to the ends of the uninterrupted gap.

For each variant, cryptic splice isoforms were clustered according to their 3' and 5' intron ends, and for each cluster, we calculated the splicing efficiency as described above for the intended spliced isoforms. Cryptic spliced isoforms were counted only for isoforms with splicing efficiency higher than 0.01.

Splicing efficiency values for all single intron variants can be found in S1 Data.

### Computing two-intron spliced isoforms ratio

For the set of two-intron variants, we needed to classify each read to one of five possible isoforms: unspliced, intron 1, intron 2, exon-skipping, or 'both introns spliced'. Reads were classified into one of these isoforms according to junctions alignment as described above. A read was classified to an isoform according to the following conditions:

- Intron 1—positive alignment to the *exon1-exon2* junction, and negative to the *exon1-intron1* and *intron1-exon2* junctions.

- Intron 2—positive alignment to the *exon2-exon3* junction, and negative to the *exon2-intron2* and *intron2-exon3* junctions.

- Exon-skipping—positive alignment to the *exon1-exon3* junction, and negative to the *exon1-intron1* and *intron2-exon3* junctions.

- Both introns—positive alignment to the concatenated *exon1-exon2-exon3* junction, and negative to all the 4 *exon-intron* junctions.

- Unspliced—negative alignment to both *exon1-exon2* and *exon2-exon3* junctions, and positive alignment to all 4 *exon-intron* junctions.

Then the splicing ratio of each isoform was determined by the ratio of its spliced isoform abundance and the total RNA abundance.

Splicing ratio values for all two-intron variants can be found in S2 Data.

### Total RNA abundance, and data filtering

Genomic DNA levels were used to determine total RNA abundance, and to filter outlier spliced isoforms.

To determine total RNA abundance, we wish to normalize by the variant's frequency in the population. Hence, total RNA abundance of variant $x$ was determined according to:

$$Total \ RNA \ abundance(x) = log_{10}\left(\frac{RNA \ frequency(x)}{DNA \ frequency(x)}\right)$$

Some RNA read alignments might be inferred as spliced isoforms due to errors in synthesis, or systematic errors in alignment. Therefore, the splicing efficiency calculation was done also on the DNA samples, and if a variant had an intended or cryptic splicing efficiency higher than 0.05 in the DNA samples the corresponding value was set to zero.

### 3'SS avoidance calculation

For each of the 11 yeast species, we examine the frequency of the 3'SS sequence motif, to check if it is avoided near introns' 3' end. First, we calculate the frequency of the two major 3'SS sequences ([C/T]AG) at positions relative to the introns' 3' end. For that purpose, in each species, we register the sequences of all the intron-containing genes at their intron's 3' end and set the end of the intron as position 0. Then, at every position downstream or upstream to the intron end, we count the number of occurrences of the two 3'SS sequences and divide it by the number of introns in each species.

Then we test if there is a statistically significant depletion of this motif at a window of 30 nucleotides upstream or downstream of the intron end. We perform the statistical analysis using sampling of random control sets. Each control set includes N random positions from coding regions in the same genome, where N is the number of introns in a species. Those positions are set as the reference positions at which we register N sequences, and measure the 3'SS

motif frequency around them. We randomly sample $10^5$ such control sets, and then we count the number of sets for which the mean frequency within a window of 30 nucleotides is lower than the mean frequency in the true introns set. *p–value* is defined according to: ($f_{motif}^{introns}$ and $f_{motif}^{control}$ are the frequency of the 3'SS motif at the true introns set, or the control set accordingly)

$$p - value(upstream) = \# \left[ \sum_{i=-32}^{-3} f_{motif}^{control} < \sum_{i=-32}^{-3} f_{motif}^{introns} \right] \cdot 10^{-5}$$

$$p - value(downstream) = \# \left[ \sum_{i=1}^{30} f_{motif}^{control} < \sum_{i=1}^{30} f_{motif}^{introns} \right] \cdot 10^{-5}$$

## A computational model for predicting single-intron splicing efficiency

We used a gradient boosting regression model to predict splicing efficiencies of library variants. The gradient boosting implementation is based on LightGbm [62] library for Python, and the feature importance inference is based on SHAP [64] library for python.

Each variant is characterized by a set of 39 features (see S2 Table). We took a set of 12,745 variants that includes all the designed single intron variants, excluding negative controls. This set was randomly divided into a training set composed of 75% of the variants and a test set with the remaining 25%. We then trained the model on the training set using 5-fold averaging cross-validation technique [63], meaning, we divided the training set to 5 subsets, each time training the model on 4 of them, using the fifth as a validation set, and predicting the splicing efficiency value for the test set. Thus, creating 5 different predictions for the test set, which we next averaged to create a single prediction.

The parameters given to the model are the following:

- Number of leaves—50

- Learning rate—0.1

- Feature fraction—0.8

- Bagging fraction—0.8

- Bagging frequency—5

- Number of boost rounds—500

- Number of early stopping rounds—5

Feature importance was inferred by running Shapley value analysis [64] on the training set for each of the 5 k-fold iterations, followed by averaging the Shapley values over the 5 iterations. Confidence interval of the mean absolute values of Shapley values was calculated by resampling the training and test set in 1000 repeats and taking the 2.5% and 97.5% quantiles of mean absolute value for each feature.

## Supporting information

**S1 Fig. Multiple barcodes, and randomized control for RNA abundance and splicing efficiency. A.** Histogram of total splicing efficiency values. Only variants with splicing efficiency $> 0$ are presented. **B.** Splicing efficiency mean variance of quartets of the same sequence design, with different barcode sequences (red arrow), compared to the distribution of mean variance of 10,000 sets of quartets, randomly chosen from the set of designs with

multiple barcodes. **C.** Pairwise Pearson correlations of total splicing efficiency values (i.e. splicing efficiency of intended + cryptic spliced isoforms) between designs with identical sequence and different 12nt barcode sequence (mean Person correlation, r = 0.76). **D.** To check if the correlation between RNA abundance and splicing efficiency results trivially from the dependence of splicing efficiency value on the total RNA abundance, we ran the same analysis as in Fig 3A on a randomized dataset. For 5,000 mock variants we randomly assign unspliced RNA levels, and spliced RNA levels. Both values are randomly chosen from a log-normal distribution. We then calculate the splicing efficiency of each mock variant, and plot the scatter of total RNA abundance and splicing efficiency. No significant correlation is observed in this mock data (p-value = 0.45).
(TIF)

**S2 Fig. 5'SS variants effect is dependent on intron length, splice sites mutations effects on splicing efficiency, and background sequence effect. A**. For two of the 5'SS variants (i.e. GUACGU, GUAUGA), the difference in splicing efficiency as described in Fig 3D, is significantly lower only for introns longer than 120 nucleotides. This figure presents the distribution of the difference in splicing efficiency for 5'SS variants after binning variants according to their intron length. **B.** Single mutation analysis of splice sites' positions. For each site, the bottom part presents the sequence logo of the consensus splice site, and above it a depiction of the effect of each mutation on the proportion of spliced variants. For each mutation, the other positions within splice sites are kept at the consensus sequence. Color bar represents the mean splicing efficiency of all variants with this mutation as a single mutation in the splice sites. Values significantly higher than zero are marked with a red circle (proportion z-test, with Bonferroni correction). **C.** Distribution of difference in splicing efficiency for variants with mutations near splice sites which result in predicted secondary structure in which the splice resides within the stem of a stem loop structure. The differences are taken compared to an unmutated variant with the same splice site sequences and length properties. For all possible combination of splice site structure mutations, the distribution of splicing efficiency is significantly lower than the splicing efficiency of the reference variant (t-test, p-value$<10^{-40}$). $\sqrt{}$ symbols mark the mutated sites. **D.** Distribution of differences in splicing efficiency for different background sequences, compared to a reference variant with the same design features and the *MUD1* background sequence. Splicing efficiency distribution for the *SNC1* background is not significantly different from *MUD1* distribution (t-test, p-value = 0.2), other background sequences have significantly lower distribution (t-test, p-value$<0.01$).
(TIF)

**S3 Fig. Intronic features effect on splicing efficiency and 3'SS sequence effect on cryptic splicing in synthetic introns. A-C.** These panels of this figure present a comparison of splicing efficiency values for all the library variants utilizing consensus splice sites' sequences. **A.** Splicing efficiency distribution is binned according to poly-pyrimidine tract strength, which is calculated as the Y (i.e. C or U) content at a window of 20 nucleotides upstream to the 3'SS. In order to specifically check elements that are not U-rich, only elements with at least 30% C out of their Y content are taken into account. Correlation is not significant (p-value = 0.79), compared to the highly significant correlation for U-rich elements (Fig 4A). **B.** Splicing efficiency distribution binned according to intronic GC content (Pearson r = -0.85 p-value$<0.01$). **C.** Splicing efficiency distribution of spliced variants for different intron lengths. **D**. Distribution of cryptic isoforms splicing efficiency for the set of synthetic introns, binned according to the original intron's 3'SS.
(TIF)

**S4 Fig. BS-to-3'SS distance distribution is associated with existence U2AF1 splicing factor.**
**A.** Distribution of BS-to-3'SS distance in each of the 11 species. Species with no copy of the gene coding for the splicing factor U2AF1 are marked in black, one species with a malfunctioned copy of U2AF1 is marked in purple, and the ones with a functional copy of U2AF1 are marked in red. **B-E.** For the same species represented in (A), and in corresponding positions, distribution of intron length (B), 5'SS-to-BS distance (C), intronic GC content (D), and poly uracil enrichment as calculated in Fig 4A. We notice that although intron length differs substantially between species with U2AF1 splicing factor, to species that lack it (B), this difference is ascribed solely to differences in BS-to-3'SS distance (A), as we see no difference in 5'SS-to-BS distance (C).
(TIF)

**S5 Fig. 3'SS motif avoidance signal for 10 yeast species.** 3' splice site motif avoidance signal for each of the other 10 yeast species that contributed introns to the library. Each panel presents the motif avoidance signal as explained in Fig 5F.
(TIF)

**S1 Table. Synthetic combinatorial design library subset features.**
(PDF)

**S2 Table. Gradient boosting model features.**
(PDF)

**S3 Table. List of primers.**
(PDF)

**S1 Data. A table with all single intron variants data, including splicing efficiency values and genetic features.**
(XLSX)

**S2 Data. A table with all double introns variants data, including splicing efficiency values and genetic features.**
(XLSX)

**S3 Data. Alignment of the endogenous *S. cerevisiae MUD1* gene to the library's *MUD1* based background gene.**
(PDF)

**S4 Data. Contingency tables for the gradient boosting model predictions Vs. measurements, and for comparison between sets of identical designs with different barcodes.**
(PDF)

## Acknowledgments

We wish to thank Sasha F. Levy for kindly providing us the yeast strain and plasmids for the Cre-Lox library integration method, and for helpful discussions on high throughput library experiments. We thank Ruth Sperling for helpful discussions on structural properties of the splicing reaction. We thank Leon Anavy and Idan Frumkin for helpful discussions on large library design, and Idan Frumkin, and Martin Mikl for critical reading of this manuscript.

## Author Contributions

**Conceptualization:** Dvir Schirman, Yitzhak Pilpel, Orna Dahan.

**Data curation:** Dvir Schirman, Zohar Yakhini, Yitzhak Pilpel, Orna Dahan.

**Formal analysis:** Dvir Schirman.

**Funding acquisition:** Zohar Yakhini, Yitzhak Pilpel.

**Investigation:** Dvir Schirman.

**Methodology:** Dvir Schirman, Zohar Yakhini, Yitzhak Pilpel, Orna Dahan.

**Project administration:** Yitzhak Pilpel.

**Resources:** Yitzhak Pilpel.

**Supervision:** Yitzhak Pilpel, Orna Dahan.

**Validation:** Yitzhak Pilpel, Orna Dahan.

**Visualization:** Dvir Schirman.

**Writing – original draft:** Dvir Schirman.

**Writing – review & editing:** Zohar Yakhini, Yitzhak Pilpel, Orna Dahan.

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
