## [Decision Letter · Decision Letter 0]

15 Jul 2021

Dear Dr Pilpel,

Thank you very much for submitting your Research Article entitled 'A broad analysis of splicing regulation in yeast using a large library of synthetic introns' to PLOS Genetics.

The manuscript was fully evaluated at the editorial level and by independent peer reviewers. The reviewers appreciated the attention to an important topic but identified some concerns that we ask you address in a revised manuscript

We therefore ask you to modify the manuscript according to the review recommendations. Your revisions should address the specific points made by each reviewer.

[LINK]

Yours sincerely,

Gil Ast

Guest Editor

PLOS Genetics

Gregory P. Copenhaver

Editor-in-Chief

PLOS Genetics

Reviewer's Responses to Questions

**Comments to the Authors:**

Reviewer #1: Schirman et al took a synthetic approach to study cis regulatory elements of alternative splicing. Many of their finding as the importance of the 3’ and 5’ SS and BS are known, but their findings might be able to fine tune our understanding of these seq.

Schirman’s results find that the most important feature is the sequence of the BS and that its importance is associated with the presence or absence of U2AF1 which binds this seq. In addition the length between the BS-to-3’SS is also of importance.

Using AI Schirman can predict 70% of splicing efficiency. Interestingly 30% of splicing events were affected by the barcode seq that was in the exon.

The manuscript is easy to read and the figures are well presented.

Major points:

• I find that the most interesting finding is that splicing efficiency is positively correlated with RNA abundance. This result should be further explored. All the library has the same promoter but does transcription initiation is indeed similar? How about elongation rate? Can the sequence change the elongation rate? Maybe the authors can use their RNAseq data to explore these options.

• The connection of BS and U2AF1 can be tested by inserting U2AF1 to cerevisiae and validating the result with a couple of transcripts.

Reviewer #2: Review is uploaded as an attachment.

Reviewer #3: Schirman and colleagues report a very detailed study of the sequence determinants of splicing in the yeast S. cerevisiae, performed using a large library of synthetic variants of a model intron. Although previous studies have addressed similar questions using related methods, this work is distinguished by its scale and by the careful experimental design that allows the authors to quantify the effects of various parameters, such as the sequence of 5'SS, BS, 3'SS, and the length and nucleotide composition of introns. A particularly interesting finding concerns the splicing efficiency of orthologous introns, and its relation to the presence and absence of the U2AF1 homologs across yeast species.

Minor comments:

Although alternative splicing is uncommon in S. cerevisiae, some endogenous yeast introns are know to be spliced more efficiently/accurately than others (PMID 29254943). It would be interesting to see whether the high-throughput measurements shown here recapitulate previous measurements of splicing efficiency and accuracy in endogenous yeast genes.

There is a risk that some of the unspliced reads represent vector or genomic DNA contamination, rather than the intended cDNA molecules. How is this mitigated?

In Figure 1 (or S1), please include an alignment of the original MUD1 sequence and the synthetic construct sequence, indicating position coordinates, cloning sites, barcodes, and other relevant sequence elements. As it stands, the design is somewhat unclear, for example how can the 5' splice site be placed in position 6-11 of the oligo (Fig 1 legend), if the random barcode is placed in position 1-12 (Fig 1B)?

Figure 2C: For clarity, the labels should read "all consensus sites", and "1 or more non-consensus sites".

Figure 3B shows a surprisingly large effect of BS-3'SS distance on splicing efficiency. For example, variants with 30-40 nt distance are almost never spliced, but other variants are spliced to various degrees. How confident are you that the effect is caused by distance per se, rather than some other sequence property (eg RNA folding or nucleotide composition)?

Why were negative control designs excluded from the analysis in the computational model shown in Fig 7?

**Have all data underlying the figures and results presented in the manuscript been provided?**

Reviewer #1: Yes

Reviewer #2: Yes

Reviewer #3: Yes

PLOS authors have the option to publish the peer review history of their article (what does this mean?). If published, this will include your full peer review and any attached files.

Reviewer #1: No

Reviewer #2: No

Reviewer #3: No

---

## [Decision Letter · Decision Letter 1]

3 Sep 2021

Dear Dr. Pilpel,

We are pleased to inform you that your manuscript entitled "A broad analysis of splicing regulation in yeast using a large library of synthetic introns" has been editorially accepted for publication in PLOS Genetics. Congratulations!

Yours sincerely,

Gil Ast

Guest Editor

PLOS Genetics

Gregory P. Copenhaver

Editor-in-Chief

PLOS Genetics

Comments from the reviewers (if applicable):

Reviewer's Responses to Questions

**Comments to the Authors:**

Reviewer #1: I feel that my suggestions would strengthen the observations in this manuscript. The authors did not. Still this is an interesting work.

Reviewer #2: The authors have responded well to our comments. They have better explained various observations including the performance of their gradient boosting ML model, the splicing efficiency of YAG introns, and the splicing efficiency of introns with 30-40 nucleotides between the branchpoint and 3’SS. They have also added a description of the effects of secondary structure around splice sites and the effects of background sequence on splicing readouts.

Reviewer #3: All my comments have been addressed, I recommend acceptance.

**Have all data underlying the figures and results presented in the manuscript been provided?**

Reviewer #1: None

Reviewer #2: Yes

Reviewer #3: None

PLOS authors have the option to publish the peer review history of their article (what does this mean?). If published, this will include your full peer review and any attached files.

Reviewer #1: No

Reviewer #2: No

Reviewer #3: No

**Data Deposition**

http://datadryad.org/submit?journalID=pgenetics&manu=PGENETICS-D-21-00763R1

**Press Queries**

---

## [Editor Report · Acceptance letter]

22 Sep 2021

PGENETICS-D-21-00763R1 

A broad analysis of splicing regulation in yeast using a large library of synthetic introns 

Dear Dr Pilpel, 

We are pleased to inform you that your manuscript entitled "A broad analysis of splicing regulation in yeast using a large library of synthetic introns" has been formally accepted for publication in PLOS Genetics! Your manuscript is now with our production department and you will be notified of the publication date in due course.

With kind regards,

Katalin Szabo

PLOS Genetics

On behalf of:
